# BatteryML: An Open-source Platform for Machine Learning on Battery Degradation

**Han Zhang**[1]*, **Xiaofan Gui**[2] , **Shun Zheng**[2], **Ziheng Lu**[2], **Yuqi Li**[3]*, **Jiang Bian**[2]

[1]Institute for Interdisciplinary Information Sciences, Tsinghua University
[2]Microsoft Research
[3]Department of Materials Science and Engineering, Stanford University
`han-zhan17@mails.tsinghua.edu.cn`,
`{xiaofangui,shun.zheng,zihenglu,jiang.bian}@microsoft.com`,
`yuqili@stanford.edu`

## Abstract

Battery degradation remains a pivotal concern in the energy storage domain, with machine learning emerging as a potent tool to drive forward insights and solutions. However, this intersection of electrochemical science and machine learning poses complex challenges. Machine learning experts often grapple with the intricacies of battery science, while battery researchers face hurdles in adapting intricate models tailored to specific datasets. Beyond this, a cohesive standard for battery degradation modeling, inclusive of data formats and evaluative benchmarks, is conspicuously absent. Recognizing these impediments, We present BatteryML[1] - a one-step, all-encompass, and open-source platform that integrates data preprocessing, feature extraction, and the implementation of both conventional and state-of-the-art models. This streamlined approach promises to enhance the practicality and efficiency of research applications. BatteryML seeks to fill this void, fostering a collaborative platform where experts from diverse specializations can contribute, thereby accelerating collective progress in battery research.

## 1 Introduction

Lithium-ion batteries, characterized by their high energy density and prolonged cycle life, have revolutionized energy storage across sectors like electric vehicles, consumer electronics, and renewable energy solutions. However, the ubiquitous adoption of these batteries comes with inherent challenges surrounding their capacity degradation and performance stability (Edge et al., 2021). Continuous cycling tends to diminish their charging and discharging capacities, posing dire implications for real-world applications. For instance, "range anxiety" becomes prevalent among electric vehicle owners, and reliability concerns arise for energy storage systems. Beyond the user experience, rapid degradation introduces broader issues, such as escalated maintenance costs, heightened resource usage, environmental strain, and potential economic inefficiencies. As such, decoding and forecasting battery performance degradation has ascended as a pivotal topic in industrial artificial intelligence.

Peeling back the layers of lithium-ion batteries reveals their intricate, non-linear electrochemical dynamics (Hu et al., 2020). Degradation, observed as diminishing performance with increased charge-discharge iterations, branches mainly into losses in lithium-ion inventory (e.g., solid electrolyte interphase film formation and electrolyte decomposition) and active material losses (e.g., graphite delamination and binder decomposition) (Pop et al., 2007; Dubarry et al., 2012; Sarasketa-Zabala et al., 2015). Moreover, the internal resistance and excessive electrolyte losses further contribute to the battery's declining health. Such losses in electrolytes, in particular, can precipitate a stark capacity plunge towards a battery's lifecycle end (Edge et al., 2021).

Confronting this degradation complexity, reliably predicting a battery's *remaining useful life* (RUL), *state of health* (SOH), and *state of charge* (SOC) becomes a herculean endeavor (Lipu et al., 2018).

---

*Han Zhang and Yuqi Li worked on this project during their internship at Microsoft Research.
[1]Project repository: `https://github.com/microsoft/BatteryML`

The significance of RUL, especially in battery management, second-hand vehicle evaluation, and more, has spurred extensive research. For instance, integrating techniques like electrochemical impedance spectroscopy with machine learning has been demonstrated to hold promise (Zhang et al., 2020; Severson et al., 2019; Attia et al., 2021; 2020). Similarly, SOH and SOC estimation has seen advances through capacity-based, Coulomb counting, impedance methods, and model-based techniques, with machine learning bringing innovative dimensions (Wang et al., 2011; Plett, 2004; Barsoukov et al., 2005; Doyle et al., 1993; He et al., 2011b).

Yet, a glaring gap persists in the domain. While individual studies have made strides in understanding battery degradation, their focal points often remain narrowly defined by specific use scenarios or charge-discharge strategies. Existing research predominantly uses particular battery types and operation paradigms, making findings less generalizable. The disparities across datasets — in terms of battery forms, chemistries, operational profiles, or environmental conditions — render a universal approach elusive. Consequently, the absence of a consistent standard in battery research underscores the need for a comprehensive and unified methodology.

**Challenges** In the realm of battery research and modeling, diverse challenges often impede the streamlined application and integration of machine learning techniques.

- **Data heterogeneity.** Battery data exhibits considerable heterogeneity in terms of both data format and data patterns. The output format of various battery testing systems differs with respect to the recorded fields, time granularity, file types, etc. Sometimes, severe conceptual confusion may arise due to differences in terminology conventions. For example, the capacity of the battery may be reported through the areal specific capacity, the total capacity, or even normalized capacity. Subsequent data processing further adds to the data heterogeneity. Even for the same data format, different cathode material compositions such as $LiCoO_2$ (LCO), $LiFePO_4$ (LFP), and $LiNiMnCoO_2$ (NMC) lead to diverse degradation patterns.

- **Domain knowledge.** On the one hand, machine learning professionals struggle to craft effective feature spaces due to the high-dimensional and heterogeneous characteristics of battery data. This intricacy presents a significant challenge in applying advanced machine learning techniques to battery performance modeling.

- **Model development.** On the other hand, battery experts, while proficient in understanding degradation mechanisms, often face challenges in building robust machine learning model due to the nuanced data cleaning, feature engineering, and model fine-tuning processes. Existing tools and models that arecrafted for specific data structures might not seamlessly adapt to other scenarios.

**Contributions** BatteryML addresses the above challenges in a holistic manner. As an inclusive open-source platform, BatteryML simplifies every stage of battery modeling, from data preprocessing and feature construction to model training and inference.

- **Unified data representation.** Recognizing the challenges of diverse battery data, BatteryML introduces a standardized data representation method. It provides comprehensive processing tools to collate and harmonize virtually all public battery datasets. With this consistent data representation, a uniform evaluation criterion for assessing battery degradation becomes feasible, promoting robust comparisons and insights across diverse battery contexts.

- **Comprehensive open-source platform.** BatteryML covers essential battery research tasks like State of Charge (SOC), State of Health (SOH), and Remaining Useful Life (RUL). It offers a holistic suite of tools encompassing data preprocessing, feature and target extraction, model training, prediction, and visualization. This integrative design allows experts from varied fields to contribute, nurturing ongoing innovation in battery research.

- **State-of-the-art model integration.** BatteryML seamlessly integrates a wide array of models, spanning both traditional and cutting-edge techniques. The platform's modular design ensures clear demarcation between models and data processing stages, facilitating effortless integration and refinement by machine learning experts. With a unified data representation, researchers can leverage multiple datasets simultaneously, unlocking techniques

like transfer learning. This fluidity not only accelerates research but also sets the stage for the integration of more sophisticated models in the coming times.

## 2 RELATED WORK

**Battery modeling tasks.** Lithium-ion battery lifetime modeling has been the subject of numerous studies. A vast number of researchers have proposed both physical and semi-empirical models to capture various mechanisms, including the growth of the solid-electrolyte interphase, lithium plating, active material loss, and impedance increase (Das et al., 2019; Palacín, 2018; Woosung et al., 2020). Predictive state estimation for remaining useful life in battery management systems often hinges upon these mechanistic and semi-empirical models. Specialized diagnostic measurements, such as coulombic efficiency and impedance spectroscopy, further assist in lifetime estimation (Burns et al., 2013; Chen et al., 2001; Tröltzsch et al., 2006; Love et al., 2014). Despite their success, these chemistry or mechanism-specific models often struggle to accurately characterize the battery degradation on the cell level due to the intricate interactions between multiple degradation modes and the thermal and mechanical variances within a cell (Waldmann et al., 2014; 2015; Bach et al., 2016; Jain et al., 2013; Aykol et al., 2016).

While semi-empirical approaches require in-depth battery and chemistry domain knowledge to model various intricate degradation mechanisms, the rapid evolution of machine learning provides a fully data-driven methodology, using linearmodels, support vector machines, and neural networks for accurate battery degradation modeling (Severson et al., 2019; Segler et al., 2018; Ng et al., 2020; Lu et al., 2023; Chemali et al., 2018; Zhang et al., 2018; Li et al., 2020; Attia et al., 2021; Ma et al., 2022; Ren et al., 2018; Khumprom & Yodo, 2019; Sahinoglu et al., 2018; Jiménez-Bermejo et al., 2018; Wu et al., 2018; Zhang et al., 2019). Such data-driven approaches allow seamlessly integration of different degradation factors such as electrical signals, temperature and electrochemical impedance spectroscopy data, for flexible battery modeling (Han et al., 2019; Li et al., 2019; Meng & Li, 2019; Liu et al., 2020; Hossain Lipu et al., 2021; Ayob et al., 2022; Rauf et al., 2022). On one hand, the flourishing of these data-driven approaches continues to propel advancements in battery data acquisition and the performance of machine learning models. On the other hand, due to the absence of a unified modeling framework, the problem settings and data representations vary across different models, making it challenging to achieve stable replication and comparison. This accentuates the urgent need for a consolidated platform that standardizes battery degradation research, further propelling the field's progression.

**Battery Early Prediction Framewok.** The Battery Evaluation and Early Prediction Software Package (BEEP) offers an open-source, Python-centric framework designed for the efficient handling and processing of extensive battery cycling data streams (Herring et al., 2020). Notable features of BEEP encompass file-system-oriented organization of raw cycling data, validation procedures for data authenticity, linear model learning for anomaly detection and cycle life early-prediction. While BEEP positions itself as a tool designed to assist battery experts with coding skills to more efficiently conduct battery life predictions to validate design ideas, BatteryML aims to bridge the gap between the battery and machine learning communities. Through its modular design, BatteryML decouples the knowledge dependencies of the two communities, enabling battery experts to utilize the most advanced machine learning models, while also allowing machine learning professionals to more effectively optimize models for battery data. BatteryML's data representation naturally supports the transformation of battery data into multidimensional tensors, thereby facilitating seamless integration for battery experts into existing deep learning frameworks. Moreover, BatteryML empowers machine learning professionals to explore advanced learning paradigms, such as transfer learning, on a variety of battery data.

## 3 BATTERYML PLATFORM

**Pipeline Overview** As depicted in Figure 1, the BatteryML pipeline comprises an organized sequence of functional modules, guiding users through the process of model creation and application. The initial step involves converting all incoming data into a consistent format. Following this, a configuration file is crafted to specify data locations, partitioning strategies, feature and label generation methods, as well as the associated model parameters. An elaborate sample of these settings

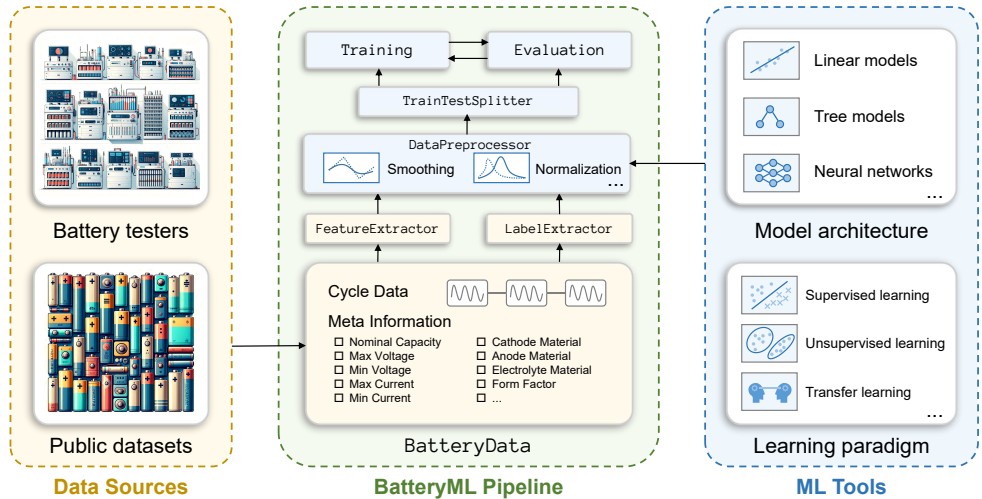

Figure 1: An overview of BatteryML.

is presented in Code 1. Once configured, the comprehensive pipeline process allows end-to-end battery degradation modeling. Components integrated in this process encompass the train-test data split module, label extractor module, feature extractor module, data preprocessing module, and the model module. Detailed examples of the pipeline's core functions, including pipeline initialization, model training, and result extraction, is available in Appendix E.1.

- **Train-test split module.** This module determines the split strategy of the learning process. BatteryML allows users to randomly allocate battery cells into training and test subsets with respect to a given split proportion. Alternatively, BatteryML also provide standard data splits for popular datasets such as MATR (Severson et al., 2019) and HUST (Ma et al., 2022) to enable reproducible and comparable experiments. The module also offers high flexibility for custom data partitioning.

- **Label extractor module.** This module automatically annotates the prediction target for major battery modeling tasks such as RUL and SOH prediction.

- **Feature extractor module.** This module implements the popular features designed by domain experts such as the cycle-difference features for battery life prediction (Severson et al., 2019). This module also enables extraction of raw electric signals and conversion to tensors for training neural networks. This module is highly extensible to support custom feature design.

- **Data preprocessing module.** This module supports flexible transformation and refinement independently for both features and labels, including data normalization and augmentation techniques to boost model performance.

- **Model module.** This module specifies the model structure and learning parameters. BatteryML currently supports a broad spectrum of machine learning models including linear models, tree-based models, and neural network-based models. Users can effortlessly manipulate the model behavior without the concerning the intricacies of the specialized domain knowledge.

## 3.1 BatteryData: A Unified Battery Data Representation

The multifaceted landscape of battery data stems from the diverse data collection apparatuses and preprocessing methodologies employed by manufacturers. This diversity results in varied data formats, terminological conventions, data fields, and signal recording strategies. Recognizing the challenge this poses, we introduced a unified data representation `BatteryData` that encompasses both meta information and charge/discharge cycles.

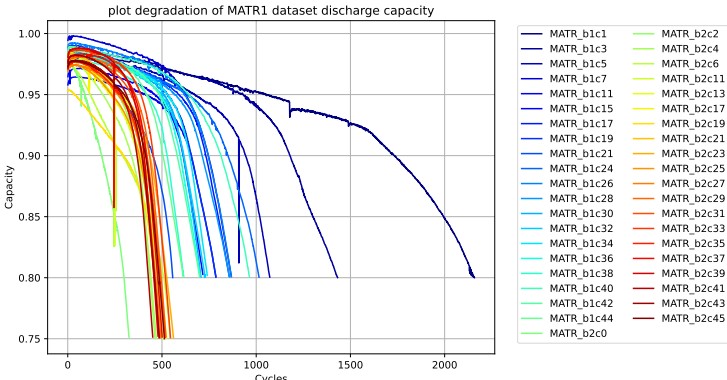

Figure 2: MATR1 degradation curve showcasing the train and test dataset discharge capacity

- **Meta information specifications.** Battery attributes such as the anode, cathode, and electrolyte materials are recorded as meta information fields. Parameters such as nominal capacity, depth of charge/discharge, and the operating thresholds for voltage and current are also summarized as cell-level attributes.
- **Charge/Discharge cycles.** `BatteryData` organizes the time series records as a list of cycles, detailing charge/discharge protocols and capacity, voltage, current, time, temperature, and internal resistance records.

A comprehensive outline of this `BatteryData` is presented in Appendix E.2.

Our commitment to this unified data representation not only facilitates efficient data management from diverse sources but also supports insightful comparisons of battery performance and highlights degradation nuances as battery ages. It creates a conducive environment for deploying machine learning and sophisticated data analysis strategies, propelling advancements in battery data operation and maintenance. For users, `BatteryData` allows flexible visualization for battery data. As evident in Figure 2, one can glean insights into capacity degradation trends by aggregating all the cycles of the battery cell. Additionally, users can delve into cycle-level signals such as the trajectory of voltage curves of successive cycles or the evolution of Coulombic efficiency for further degradation analysis, as highlighted in Figure 3. Such graphical interpretations provide users with clearer and more intuitive understanding of battery performance and degradation pathways, which in turn benefits the feature and model design.

BatteryML seamlessly supports converting output formats from various battery cycler systems into `BatteryData` out of the box. Moreover, we offer a suite of automated processing tools to transform existing datasets into the `BatteryData` format. These dual capabilities ensure BatteryML's excellent compatibility with a wide range of battery data sources.

## 3.2 FEATURE ENGINEERING

BatteryML boasts an array of degradation features, enabling users to flexibly tailor the data based on distinct experimental needs. These features bifurcate into two categories: within-cycle and between-cycle features.

**Within-cycle features.** This category encompasses characteristics observed within individual charge/discharge cycles. Examples include

- *QdLinear*, which is derived by linear interpolation of the capacity-voltage curve in discharge cycles (Attia et al., 2021).
- *Coulombic efficiency*, which is the fraction of charge and discharge capacity within a single cycle, an important indicator of how efficiently a battery can release its stored energy.
- *Internal resistance*, which can be calculated by the voltage and current signals to measure the opposition to the flow of electric current within the battery.

**Between-cycle features.** These features captures the degradation patterns of any battery cell on a higher level, usually across multiple cycles. Examples include

- *Variance of the difference of QdLinear curves*, an intuitive yet effective feature that indicates battery degradation speed (Severson et al., 2019).
- *Capacity decay dynamics*, the slope of capacity decay curve fitted in early cycles.
- *Average charging time*, which reflects the irreversible structural changes within the battery, such as lithium plating and the growth of the SEI layer.
- *Temperature dynamics*, which indicates the intensity of the electrochemical reactions occurring within the battery.
- *Minimal internal resistance*, reflecting the upper bound of battery health state.

Both the within- and between-cycle features are represented as tensors that are compatible with modern machine learning frameworks. This allows a flexible manipulation of the feature space and a natural combination with deep learning models. We provide a detailed introduction of BatteryML feature module in Appendix E.3. Through these feature extraction methodologies, BatteryML empowers users to freely design and combine features, significantly simplifying the process of reproducing existing work and testing new features.

### 3.3 AUTOMATIC LABEL ANNOTATION

BatteryML supports automatic label annotation for supervised battery degradation modeling. For different degradation modeling tasks, BatteryML calculates the label as tensors according to the definition. Here we briefly introduce the most important degradation modeling tasks.

As batteries undergo continuous cycles, their capacity inevitably declines due to aging, resulting in slight variances in performance with each use. From the moment of manufacture until a battery is fully aged and retired, a direct and crucial question is to measure the loss of the battery's maximum available capacity in any given cycle relative to its capacity when new, namely, the cycle's State of Health (SOH). Specifically, if we denote the battery's nominal capacity as $C_{\text{nom}}$ and the discharge capacity in the current cycle as $C_{\text{full}}$, the SOH is defined as the percentage of these two quantities

$$\text{SOH} = \frac{C_{\text{full}}}{C_{\text{nom}}} \times 100\%. \tag{1}$$

Here the $C_{\text{full}}$ in a strict definition of SOH refers to the capacity measured under the same discharge protocol as the nominal capacity. Note that in practice, this requires additional cycles under constrained temperature and current conditions and is usually infeasible due to significant human labor costs.

This requires accurately estimating the ratio of the capacity already discharged in the current cycle to the original total capacity, that is, the State of Charge (SOC). Essentially, when the depth of discharge is 100%, and the Battery Management System (BMS) has recorded the discharged capacity, SOC is equivalent to SOH. Estimating SOC becomes more challenging when the depth of discharge is less than 100% (for example, when discharging begins again after a brief charge). Specifically, denoting the remaining capacity of the battery as $C_{\text{curr}}$, the definition of SOC is

$$\text{SOC} = \frac{C_{\text{curr}}}{C_{\text{full}}} \times 100\%. \tag{2}$$

SOH requires a prediction for each cycle, whereas SOC demands a prediction at every moment during charging and discharging phases. Both tasks requires online prediction, with SOC estimation imposing higher demands on the real-time performance of the model. Another offline task, early battery life prediction, involves conducting a limited number of charge-discharge tests to predict the full lifespan of the battery—typically until its capacity falls below 80% of the nominal capacity (Li et al., 2019)—before significant degradation occurs. This task can greatly shorten the cycling test duration for batteries, playing a crucial role in downstream battery optimization tasks such as the selection of battery materials, optimization of charging strategies, and control of operating temperatures.

BatteryML automates the task of label annotation through sequential traversal of each cycle's charge/discharge stages, significantly reducing the reliance of model design on battery domain knowledge and allowing machine learning experts to focus on developing superior models.

## 3.4 MODEL DEVELOPMENT

Following (Attia et al., 2021), BatteryML incorporates multiple off-the-shelf baselines for battery lifetime predictions, including linear models, tree-based models and neural networks. Domain-Enhanced methods like the 'Variance', 'Discharge', and 'Full' models from the (Severson et al., 2019) are implemented using handcrafted features and linear models. BatteryML also includes statistical models such as Ridge regression (Hoerl & Kennard, 2000), Principal Component Regression(PCR) (Tipping & Bishop, 1999), Partial Least Squares Regression(PLSR) (Geladi & Kowalski, 1986), Gaussian process (Williams & Rasmussen, 2006), XGBoost (Chen & Guestrin, 2016), Random forest (Breiman, 2001) models. For high-performance needs, we offer neural network models like Multi-Layer Perceptron (MLP) (Haykin, 1994), Convolutional Neural Networks (CNN) (Krizhevsky et al., 2012), Long Short-Term Memory networks (LSTM) (Hochreiter & Schmidhuber, 1997). Additionally, we introduce Transformer (Vaswani et al., 2017), a ground-breaking architecture in language and vision domains (Brown et al., 2020; Dosovitskiy et al., 2021), as a new neural baseline. These implementations utilize scikit-learn (Pedregosa et al., 2011) for all statistical models, barring XGBoost, and PyTorch (Paszke et al., 2019) for neural networks. We re-train each model with 10 random seeds and report averaged results to eliminate the effect of random initializaiton.

To cater to the diversity of experimental contexts, users have the liberty to tweak and customize these models as per their needs. A deeper dive into model intricacies can be found in Appendix B.

BatteryML's versatility lies in offering a spectrum of models, ensuring users can cherry-pick and tailor the most fitting analytical approach aligned with their research objectives. Anchored on `BatteryData`, BatteryML paves the way for integrating cutting-edge machine learning paradigms like transfer learning and multi-task learning into battery modeling Moreover, as we sail through an era where large-scale model architectures are blossoming, BatteryML lays a robust foundation to harness the power of these expansive models for future battery research.

## 4 EVALUATION

In this section, we provide an in-depth evaluation of model performance across various datasets to inform model selection. Through a comprehensive analysis, our intent is to offer a holistic perspective on the efficacy of each model, empowering researchers and practitioners to make informed decisions tailored to their specific goals.

## 4.1 DATA

We based our evaluation on several publicly accessible battery datasets: CALCE (Xing et al., 2013; He et al., 2011a), HNEI (Devie et al., 2018), HUST (Ma et al., 2022), MATR (Severson et al., 2019; Hong et al., 2020), RWTH (Li et al., 2021), SNL (Preger et al., 2020), and UL_PUR (Juarez-Robles et al., 2020; 2021). These datasets encompass LFP, LCO, NMC, NCA and NMC_LCO battery types. Further details are outlined in Table 1. Certain datasets were excluded due to their unsuitability for tasks such as RUL estimation. The datasets differ in terms of materials, capacities, voltages, and RUL ranges. For RUL tasks, we also created combined datasets from the public sources to assess training efficacy when various battery data are combined. Notably, CRUH combines CALCE, RWTH, UL_PUR, and HNEI datasets; CRUSH merges CALCE, RWTH, UL_PUR, SNL, and HNEI datasets; and MIX incorporates all datasets used in our study. For more detailed information on the data, please refer to the Appendix A.

## 4.2 BATTERY DEGRADATION MODELING

BatteryML currently supports battery degradation tasks, including RUL prediction, SOH estimation and SOC estimation. Here we report the main benchmark results, and leave the detailed analysis and further ablation studies in the appendix.

Table 1: Specifications of data sources.

| Data source | Electrode chemistry | Nominal capacity | Voltage range (V) | RUL dist. | SOC dist. (%) | SOH dist. (%) | Cell count |
|---|---|---|---|---|---|---|---|
| CALCE | LCO/graphite | 1.1 | 2.7-4.2 | 566±106 | 77±17 | 48±30 | 13 |
| MATR | LFP/graphite | 1.1 | 2.0-3.6 | 823±368 | 93±7 | 36±36 | 180 |
| HUST | LFP/graphite | 1.1 | 2.0-3.6 | 1899±389 | 100±10 | 43±28 | 77 |
| HNEI | NMC_LCO/graphite | 2.8 | 3.0-4.3 | 248±15 | 64±17 | 49±28 | 14 |
| RWTH | NMC/carbon | 1.11 | 3.5-3.9 | 658±64 | 60±24 | 46±22 | 48 |
| SNL | NCA,NMC,LFP/graphite | 1.1 | 2.0-3.6 | 1256±1321 | 86±7 | 45±27 | 61 |
| UL_PUR | NCA/graphite | 3.4 | 2.7-4.2 | 209±50 | 89±6 | 41±33 | 10 |

Table 2: Benchmark results for remaining useful life prediction. The comparison methods are split into four types, including 1) dummy regressor, a trivial baseline that uses the mean of training label as predictions; 2) linear models with features designed by domain experts; 3) traditional statistical models with *QdLinear* feature; 4) deep models with *QdLinear* feature. For models sensitive to initialization, we present the error mean across ten seeds and attach the standard deviation as subscript.

| **Models** | MATR1 | MATR2 | HUST | SNL | CLO | CRUH | CRUSH | MIX |
|---|---|---|---|---|---|---|---|---|
| Dummy regressor | 398 | 510 | 419 | 466 | 331 | 239 | 576 | 573 |
| "Variance" model | 136 | 211 | 398 | 360 | 179 | 118 | 506 | 521 |
| "Discharge" model | 329 | **149** | **322** | 267 | 143 | 76 | >1000 | >1000 |
| "Full" model | 167 | >1000 | 335 | 433 | **138** | 93 | >1000 | 331 |
| Ridge regression | 116 | 184 | >1000 | 242 | 169 | 65 | >1000 | 372 |
| PCR | **90** | 187 | 435 | **200** | 197 | 68 | 560 | 376 |
| PLSR | 104 | 181 | 431 | 242 | 176 | **60** | 535 | 383 |
| Gaussian process | 154 | 224 | >1000 | 251 | 204 | 115 | >1000 | 573 |
| XGBoost | 334 | 799 | 395 | 547 | 215 | 119 | **330** | 205 |
| Random forest | $168_9$ | $233_7$ | $368_7$ | $532_{25}$ | $192_2$ | $81_1$ | $416_5$ | **$197_0$** |
| MLP | $149_3$ | $275_{27}$ | $459_9$ | $370_{81}$ | $146_5$ | $103_4$ | $565_9$ | $451_{42}$ |
| CNN | $102_{94}$ | $228_{104}$ | $465_{75}$ | $924_{267}$ | >1000 | $174_{92}$ | $545_{11}$ | $272_{101}$ |
| LSTM | $119_{11}$ | $219_{33}$ | $443_{29}$ | $539_{40}$ | $222_{12}$ | $105_{10}$ | $519_{39}$ | $268_9$ |
| Transformer | $135_{13}$ | $364_{25}$ | $391_{11}$ | $424_{23}$ | $187_{14}$ | $81_8$ | $550_{21}$ | $271_{16}$ |

**Remaining useful life prediction.** In the task of RUL prediction, BatteryML models predict the number of cycles until a battery's SOH falls below a certain threshold, e.g. 80%, in comparison with the nominal capacity. The performance metrics of various methods are presented in Table 2.

Linear models using handcrafted features, such as the "Discharge" and "Full" model, offer relatively accurate predictions for LFP battery datasets. However, their performance diminishes on the CRUSH and MIX dataset, which features diverse aging conditions, due to the limited feature set and model capacity.

Traditional statistical models, capable of discerning non-linear patterns from low-level features such as $Q_d(V_d)$ curves, employ specific modeling mechanisms such as the decision tree ensemble approach in Random Forests and XGBoost. Despite robust performance on CRUH, CRUSH, and MIX, their efficacy decreases on datasets such as MATR2 and SNL, where the number of training samples are limited. This finding indicates that these statistical models require a larger volume of training data to effectively learn and represent meaningful insights in RUL task.

Neural network models, through automatic representation learning on low-level features, offer advancements, but face significant performance variations due to different random parameter initializations. For instance, our observations of CNN reveal its ability to make accurate predictions with many random seeds (as exemplified by the results on MATR1, see Table C.1). However, certain seeds can lead to a surprising increase in error, causing significant regression error variations. This underscores both the potential benefits and challenges of applying neural networks to RUL prediction

tasks. The observed disparities in performance across various network architectures also highlight the absence of a universally optimal architecture for battery modeling.

From the feature space perspective, linear models, utilizing handcrafted features, have demonstrated satisfactory performance on datasets such as `MATR2`, `HUST`, and `CLO`, which solely consist of one battery type, LiFePO4 (LFP). This finding validates the efficacy of domain knowledge. However, these models appear to be less successful when applied to datasets that encompass a wider range of battery types and aging conditions, such as `CRUSH` and `MIX`. In these instances, models that are directly fitted on the `Qd(Vd)` curve have proven to be more effective than those using manually crafted features. This highlights a deficiency in domain-specific feature design and underscores the necessity for more versatile, generalizable features, emphasizing the potential advantages of automated representation learning.

Please refer to the appendix C.1 for more detailed comparison analysis. We also provided an in-depth exploration of the impact of features and model hyperparameters in appendix D.

**State of Health estimation.** SOH estimation task requires model to predict the ratio of the current discharge capacity in reference performance test (RPT) to the nominal capacity. Since the RPT results are not always available in the public datasets, we turn to predict the ratio of observed discharge capacity to nominal capacity in this study[2]. We directly employ cells from the data sources in Table 1 for training and evaluation. Table C.2 showcases the comparison results.

The effectiveness of methods in SOH prediction varies across datasets. Linear models are generally effective but face challenges with the `MATR` cells due to variable charging strategies. Tree-based models show consistent, robust performance across datasets, establishing a strong baseline in SOH estimation. Deep learning models, however, haven't consistently outperformed traditional methods, indicating potential areas for improvement. We provided detailed analysis in appendix C.2.

**State of Charge estimation.** Similar to SOH estimation, the exact SOC value is unattainable in practice by definition. Given the fact that RPT results are also not available in most public datasets, in this study we predict the SOC derived from the observed discharge capacity. Table C.2 demonstrates the benchmark results.

LightGBM consistently surpassed other methodologies in most tasks, thereby establishing tree-based models as the current state-of-the-art in SOC prediction. Moreover, linear models continue to excel over deep learning models, highlighting the need for further research to unlock the full potential of neural networks in battery modeling. For detailed insights, please see the appendix C.3.

## 5 CONCLUSION

At the core of BatteryML is a commitment to fostering collaboration and bridging divides. As a comprehensive open-source platform, it effectively bridges the knowledge chasm between battery researchers and AI experts, streamlining data preprocessing, feature extraction, and model application, both traditional and advanced. This synthesis not only elevates battery modeling endeavors but also catalyzes a two-way exchange, i.e., empowering battery scientists to harness AI-driven tools for research and equipping AI experts with insights to tackle intricacies specific to the battery sector.

Furthermore, BatteryML serves as an anchor in standardizing practices within the battery research realm. By pioneering a unified data format and integrating advanced models into the baseline, BatteryML promotes consistency and rigour, thereby catalyzing a harmonious evolution of the industry. It is our aspiration that through BatteryML, the pace of research in battery degradation is accelerated, fostering seamless integration across industry, academia, and research spheres.

In the future, we envision BatteryML will be developed to facilitate the translation of lab data into tangible real-world applications. Such advancements promise to bolster battery research, propelling us closer to a sustainable future. Moreover, there lies an opportunity to render the platform even more user-friendly. By integrating features like one-click battery life prediction and rolling out an intuitive user interface, BatteryML can resonate with, and cater to, an even wider audience.

---

[2]BatteryML can effectively construct more accurate label for training when RPT results are available.

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

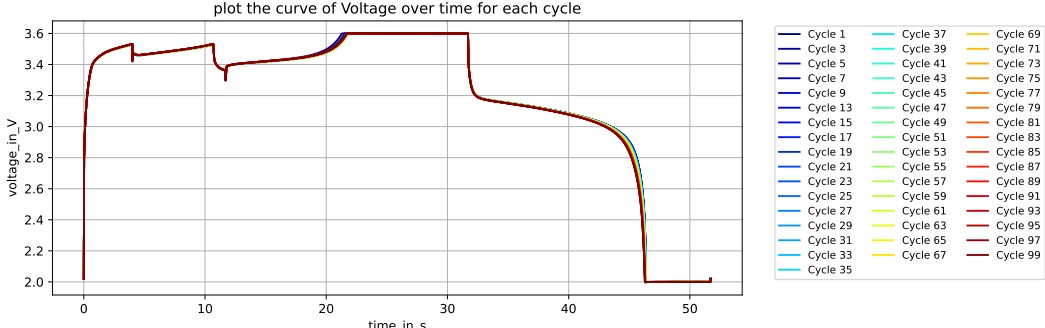

Figure 3: Voltage curves of a battery in `MATR1`. Each curve corresponds to a cycle.

## A    BENCHMARK DATASET CONSTRUCTION

Table 1 summarizes the data sources utilized in this paper. Herein, we provide a detailed introduction to these data sources and elaborate on the construction process of our benchmark datasets.

**CALCE data source (Xing et al., 2013; He et al., 2011a)**   The CALCE data source is derived from the CS2 and CX2 batteries with full lifecycle data, released by the center for advanced life cycle engineering. All the batteries in this data source are prismatic with a Lithium Cobalt Oxide (LCO) cathode. Additionally, Energy Dispersive Spectroscopy (EDS) results revealed trace elements of Manganese in these batteries. Their nominal capacity are 1100 mA. All cells in the dataset were subjected to the same charging profile, which followed a standard constant current/constant voltage protocol. This involved charging at a constant current rate of 0.5C until the voltage reached 4.2V, followed by maintaining 4.2V until the charging current decreased to below 0.05A. The cutoff voltage for these batteries was set at 2.7V.

**MATR data source (Severson et al., 2019; Attia et al., 2020)**   The MATR dataset comprises two groups of commercial 18650 Lithium Iron Phosphate (LiFePO$_4$, LFP) batteries, provided respectively by (Severson et al., 2019) and (Attia et al., 2020). It consists of 180 cells and represents the largest publicly available dataset containing complete charge-discharge cycles of batteries. These batteries were cycled in horizontal cylindrical fixtures on a 48-channel Arbin LBT potentiostat, situated within a forced convection temperature chamber maintained at 30°C. The cells have a nominal capacity of 1.1 Ah and a nominal voltage of 3.3 V. They were subjected to the same discharge strategy but different fast charging strategies, until their discharge capacity decreased to 80% of the rated capacity. In our benchmark study, a total of 180 cells are distributed across three distinct datasets: MATR1, MATR2, and CLO. This categorization is a result of the cells being measured in distinct batches.The MATR1 and MATR2 datasets are provided by (Severson et al., 2019) . The third dataset, CLO, is provided by (Attia et al., 2020).

**HUST data source (Ma et al., 2022)**   The HUST data source consists of 77 Lithium Iron Phosphate (LFP) batteries, identical in model to those used in the MATR dataset. These cells were subjected to an identical charging protocol but employed different multi-stage discharge protocols, all conducted at a constant temperature of 30°C.

**HNEI data source (Devie et al., 2018)**   This data source comprises commercial 18650 cells that feature a graphite negative electrode and a blended positive electrode composed of Nickel Manganese Cobalt (NMC) and Lithium Cobalt Oxide (LCO). These cells were cycled at a rate of 1.5C to 100% Depth of Discharge (DOD) for over 1000 cycles, conducted at room temperature.

**SNL data source (Preger et al., 2020)**   This data source includes commercial 18650 cells made of Nickel Cobalt Aluminum (NCA), Nickel Manganese Cobalt (NMC), and Lithium Iron Phosphate (LFP) materials. These cells are cycled to 80% capacity, with the cycling process still ongoing. The study focuses on evaluating the impact of temperature, Depth of Discharge (DOD), and discharge

current on the long-term degradation of these commercial cells. Each cycling round involves a capacity check, a set number of cycles under specific conditions designated for each cell, followed by another capacity check at the end. The capacity check entails three charge/discharge cycles ranging from 0 to 100% State of Charge (SOC) at a 0.5C rate.

**UL_PUR data source (Juarez-Robles et al., 2020; 2021)**  The data source consists of commercial pouch cells featuring a graphite negative electrode and a Nickel Cobalt Aluminum (NCA) positive electrode. These cells were cycled at a rate of 1C within a voltage range of 2.7-4.2V, corresponding to 0-100% State of Charge (SOC), at room temperature. This cycling was conducted until various levels of capacity fade, ranging from 10-20%, were reached. Additionally, the modules were cycled at a rate of C/2 within a voltage range of 13.7-21.0V, also corresponding to 0-100% SOC, at room temperature, until they reached 20% capacity fade.

**RWTH data source (Li et al., 2021)**  This data source comprises cycling records of 48 lithium-ion battery cells. All 48 cells, being of the same type, underwent aging under identical profiles and conditions. Before initiating the aging process, the initial performance of the cells was determined through a comprehensive Begin-Of-Life (BOL) test. Regularly scheduled Aging Reference Parameter Tests (RPT) were performed to assess the current performance of the cells. The specific cells used are Sanyo/Panasonic UR18650E cylindrical cells. These are commercially available and are produced in large quantities using an established fabrication process. The design of these cells includes a carbon anode and a Nickel Manganese Cobalt (NMC) cathode material.

**Benchmark dataset synthethis.**  We follow the original train-test split in (Severson et al., 2019) to obtain two datasets `MATR1` and `MATR2` from `MATR` data source. Notably, the charging policies of the train and test cells in `MATR1` are identically distributed, while in `MATR2` the charging policies of the test cells are unseen. We also create a `CLO` dataset that randomly split the cells from the entire `MATR` data source. We also follow the original train-test split in the `HUST` data source to obtain the `HUST` dataset for evaluation. These four datasets employ the same battery model.

We combine the cells in `CALCE`, `RWTH`, `UL_PUR`, and `HNEI` data sources into a dataset `CRUH` as the cell count of these four data sources are limited. For the cells in `SNL` data source, we first construct a dataset `SNL` by random train-test split. Then, to fully utilize the batteries with cycle life smaller than 100, we combine them with `CALCE`, `RWTH`, `UL_PUR`, and `HNEI` data sources to obtain `CRUSH`. In `CRUSH`, the models are required to predict the 90% state of health point with only the first 20 cycles.

Finally, we combine the cells in all data sources to obtain the `MIX` dataset, which to the best of our knowledge, is the largest battery degradation dataset with complete cycling records.

## B  Detailed Model Description

To facilitate the replication of existing studies, BatteryML offers a range of model implementations, encompassing both traditional statistical models and neural networks. Below, we provide a summary of the currently available models. It is important to note that our repertoire of models is continually expanding.

**Dummy Regression.**  This model uses the mean of the training samples as predictions, usually acting as a trivial baseline to indicate the error upper bound. The `DummyRegressor` from the `scikit-learn` (Pedregosa et al., 2011) toolkit is adopted in BatteryML.

**Linear Regression.**  A statistical method for modeling the relationship between a dependent variable and one or more independent variables using a linear equation. BatteryML utilizes the `LinearRegression` from `scikit-learn` to support replication of linear baselines, with default parameters applied in our experiments.

**Elastic Net (Zou & Hastie, 2005)**  A regularized regression method that linearly combines the L1 and L2 penalties of the Lasso and Ridge methods, useful for models with many correlated features. The `ElasticNetCV` from the `scikit-learn` toolkit is adopted in BatteryML.

**Gaussian Process (Williams & Rasmussen, 2006)**    A non-parametric approach in machine learning where the data is assumed to follow a Gaussian process, allowing for predictions with uncertainty estimation. BatteryML employs the `GaussianProcessRegressor` from the `scikit-learn` toolkit, providing a probabilistic model for non-linear regression tasks.

**Principal Component Regression (Tipping & Bishop, 1999)**    PCR combines principal component analysis (PCA) for dimensionality reduction with linear regression, focusing the regression on the most significant data variations. Battery utilizes a pipeline of PCA and linear regression in `scikit-learn` for implementation of PCR.

**Partial Least Squares Regression (Geladi & Kowalski, 1986)**    PLSR, implemented using the `PLSRegression` from the `scikit-learn` toolkit, is applied for modeling relationships between input and output variables. It is a statistical method that projects both the predictive variables and the response variables to a new space to improve the prediction accuracy.

**Ridge Regression (Hoerl & Kennard, 2000)**    BatteryML incorporates the `scikit-learn` implementation of Ridge regression, a method of estimating the coefficients of multiple-regression models in scenarios where independent variables are highly correlated, based on L2 regularization.

**Support Vector Regression (Bishop, 2006)**    SVR is an extension of support vector machines (SVMs) that supports regression, fitting the best line within a threshold value to the data. We still adopts the implementation of `scikit-learn`.

**XGBoost Regression (Chen & Guestrin, 2016)**    XGBoostRegressor is an implementation of the gradient boosting decision trees algorithm, designed for speed and performance, that is particularly suited for regression problems, providing a robust and versatile model capable of handling a variety of data types, achieving excellent predictive performance.

**LightGBM Regression (Ke et al., 2017)**    LightGBMRegressor is an advanced implementation of gradient boosting decision tree algorithm, developed by Microsoft, that is designed for efficiency of compute time and memory resources. It is especially suitable for handling large datasets, excels in regression problems and offers superior predictive performance. LightGBM supports both continuous and categorical features.

**Random Forest (Breiman, 2001; Geurts et al., 2006)**    Random forest regression is an ensemble learning method that operates by constructing multiple decision trees at training time to improve the predictive accuracy and control over-fitting. BatteryML incorporates RF as a strong tree-based baseline for benchmarks.

**Multilayer Perceptron (Haykin, 1994)**    A class of feedforward artificial neural network that consists of multiple layers of nodes, each layer fully connected to the next one. Please refer to our configuration files for detailed specifications.

**Long-Short-Term Memory (Hochreiter & Schmidhuber, 1997)**    This is a type of recurrent neural network (RNN) capable of learning long-term dependencies, widely used in sequence prediction problems. BatteryML features a LSTM network layer followed by a linear modeling layer to fit the sequential records of the cycles.

**Gated Recurrent Unit (Cho et al., 2014)**    Gated Recurrent Unit (GRU) is a variant of the recurrent neural network (RNN), designed to adaptively capture dependencies of different time scales. Like the LSTM, GRU is also used extensively in sequence prediction problems. It features gating units that modulate the flow of information inside the unit, without having separate memory cells. They have been shown to perform comparably to LSTM with less complexity and computational cost.

**Convolutional Neural Networks (Krizhevsky et al., 2012)**    A deep learning algorithm which can take in an input image, assign importance to various aspects/objects in the image and differentiate one from the other. In BatteryML we view the number of cycles and the interpolation dimension as the height and width of an image. Please refer to our implementations for details.

**Transformer (Vaswani et al., 2017)**   A deep learning model that adopts the mechanism of attention, differentially weighting the significance of different parts of the input data, highly effective in handling sequential data like text and time series.

## C   DETAILED COMPARISON ANALYSIS

In this section, we benchmark and compare the effectiveness of various machine learning models in three typical battery aging modeling applications, providing a detailed comparison of different methodologies available for degradation prediction.

### C.1   REMAINING USEFUL LIFE PREDICTION

Table 2 presents the performance of various methods on the task of predicting the useful life of batteries. These methods are primarily divided into four categories. The first category, Dummy Regressor, is a baseline using the mean of the training label for prediction. The second category, including the Variance, Discharge, and Full models, comprises linear regression models based on features designed by experts in the battery domain. The Variance model considers only the variance of the *QdLinear* feature curve. The Discharge model takes into account characteristics during the discharge period, while the Full model also incorporates temperature and charging features. Note that the Full model does not completely encompass the features of Discharge model. For more detailed methodologies on feature construction, refer to the (Severson et al., 2019; Attia et al., 2021) our code implementation. The third category consists of traditional statistical models, including linear methods and tree models, which utilize the original *QdLinear* curves calculated by the difference of 100-th and 10-th cycles as their input feature space. The fourth category involves neural network models, with inputs being the *QdLinear* curves from cycle 1 to 100, minus the data from cycle 10.

**Linear Models with Hand-Crafted Features.**   The Variance model, which uses the variance of the incremental `Qd(Vd)` curve as a scalar feature, displays moderate prediction accuracy across various datasets but generally falls short when compared to the Discharge and Full models. Both the Discharge and Full models exhibit variable effectiveness with significant errors on certain datasets, indicating a need for enhancements in feature design to improve the accuracy of linear model fitting.

**Traditional Statistical Models.**   These models achieve significant results by fitting on low-level features, such as raw `QdLinear` curves. Ridge Regression displays consistent, albeit modest, performance across datasets, affirming a strong linear correlation between `Qd(Vd)` feature and battery cycle life. However, its underperformance on datasets like `HUST` and `CRUSH`, underscores the challenges in linear correlation learning. Principal Component Regression(PCR and artial Least Squares Regression(PLSR), which apply principal component analysis and project cells into different subspaces before fitting a linear regression, excel in `MATR1`, `CRUH`, and `SNL` datasets. This suggests that the `Qd(Vd)` feature exhibits a linear relationship in different subspaces with the battery life. Gaussian Process Regression (GPR) underperforms PCR and PLSR on all datasets, indicating that the pairwise kernel functions struggle to capture the effective degradation patterns for all kinds of batteries. XGBoost and Random Forest, exhibiting strong performance on `CRUSH` and `MIX` datasets, indicate that tree-based models' non-linear capabilities are adept at effectively modeling complex battery aging patterns.

**Neural Network Models.**   Neural network models, including Multilayer Perceptron (MLP), Convolutional Neural Network (CNN), Long Short-Term Memory (LSTM), and Transformer architectures, display significant performance variability. MLP performs well on datasets such as `SNL`, `CLO`, and `CRUH`, while CNN shows high sensitivity to initial conditions. LSTM, exhibiting less variability, proves its robustness and superior performance on the `MATR2` dataset among the neural network methods.

In summary, there is no universally optimal method for battery modeling; all methods only excel in certain datasets and may exhibit error divergence in others. This indicates that considerable scope exists for improvement in the accuracy of early battery life prediction. Moreover, existing methods require targeted enhancements to adapt to the complex aging patterns of batteries.

Table 3: Evaluation Result of CNN with many random seeds for RUL task

| Method | seed0 | seed1 | seed2 | seed3 | seed4 | seed5 | seed6 | seed7 | seed8 | seed9 |
|--------|-------|-------|-------|-------|-------|-------|-------|-------|-------|-------|
| | | | | | Test RMSE | | | | | |
| CNN | 76 | 67 | 64 | 74 | 60 | 82 | 65 | 79 | 367 | 78 |

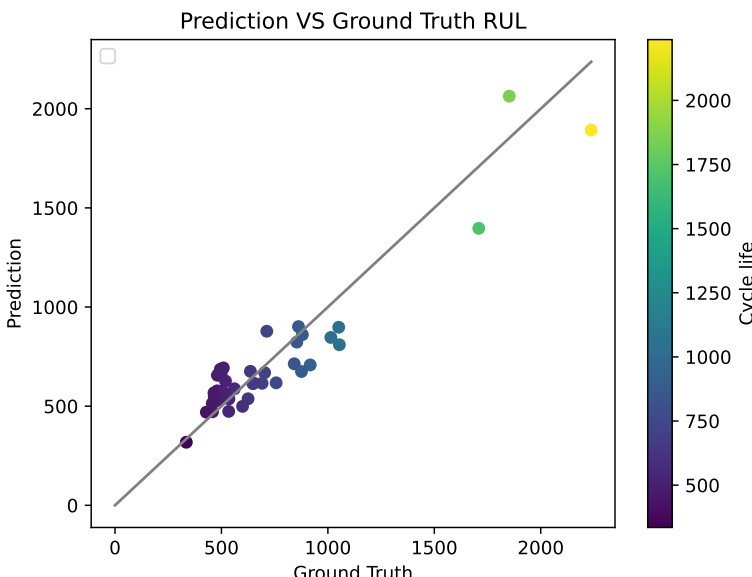

Figure 4: A showcase of the correlation between actual and predicted values in the RUL task.

## C.2  STATE OF HEALTH ESTIMATION

In practical applications, the estimation of a battery's State of Health (SOH) necessitates the prediction of the current discharge capacity under standardized conditions through reference performance test (RPT), utilizing the charging signal of the present cycle along with historical cycling data. However, the disparity between real-world battery discharge workloads and standard conditions often poses a complication in obtaining precise ground-truth labels, a persistent hurdle in the domain of battery aging modeling. To counter this, we approximate the prediction of discharge capacity under current conditions.

The SOH in this experiment is quantified as a percentage, representing the ratio of a battery's maximum discharge capacity to its nominal capacity per cycle following the Formula 1. For example, a battery with a nominal capacity of 2Ah and a current maximum discharge capacity of 1.8Ah equates to an SOH of $\frac{1.8}{2} \times 100 = 90\%$, yielding an experimental SOH range of 0 to 100.

This task leverages datasets from an array of sources, including CALCE, HNEI, HUST, MATR, RWTH, SNL, and UL_PUR. Feature extraction is comprehensive, encompassing elements such as the ratio of the current cycle's maximum charge capacity to the nominal capacity, and the voltage of the initial cycle. We scrutinized a variety of models, including Random Forest, Linear Regression, Ridge Regression, and Neural Networks like GRU, MLP, and LSTM. Each model was run with 10 different seeds on specific data, so the values in each cell of the table represent the mean and standard deviation (std) of the RMSE results from multiple seeds. The benchmark results of the SOH estimation task are delineated in Table C.2.

The efficacy of various methods demonstrates significant variability across different datasets. Linear models generally exhibit proficient prediction of the SOH across the majority of datasets but underperform when applied to the MATR cells. This suboptimal performance is largely attributed to considerable fluctuations in charging strategies among batteries within the MATR data source.

Table 4: Evaluation Result for SOH (State of Health) task

| Method | Test RMSE | | | |
|---|---|---|---|---|
| | CALCE | HNEI | HUST | MATR |
| Linear Reg | **0.45 ± 0.06** | **0.30 ± 0.01** | 4.74 ± 4.84 | 252.75 ± 430.58 |
| Ridge Reg | 0.46 ± 0.06 | 0.31 ± 0.01 | 4.62 ± 4.68 | 255.27 ± 434.10 |
| PLSR | 0.60 ± 0.10 | 0.36 ± 0.01 | 4.39 ± 4.39 | 258.93 ± 441.25 |
| PCR | 3.93 ± 9.22 | 0.52 ± 0.02 | **4.00 ± 4.23** | 755.23 ± 1219.55 |
| Random Forest | 0.72 ± 0.36 | 0.38 ± 0.04 | 6.04 ± 4.49 | **0.53 ± 0.30** |
| LightGBM | 0.74 ± 0.33 | 0.34 ± 0.02 | 4.30 ± 3.61 | 0.97 ± 0.44 |
| LSTM | 16.78 ± 1.41 | 16.90 ± 0.23 | 9.55 ± 1.92 | 1.33 ± 0.59 |
| MLP | 16.73 ± 2.00 | 14.36 ± 0.86 | 8.62 ± 2.05 | 2.89 ± 0.72 |
| GRU | 16.77 ± 1.41 | 16.88 ± 0.23 | 9.25 ± 1.78 | 1.43 ± 0.45 |

| Method | Test RMSE | | |
|---|---|---|---|
| | RWTH | SNL | UL_PUR |
| Linear Reg | 26.48 ± 81.96 | 2.89 ± 2.92 | **0.75 ± 0.05** |
| Ridge Reg | 15.68 ± 47.79 | 2.90 ± 2.93 | **0.75 ± 0.05** |
| PLSR | 11.83 ± 33.53 | 2.77 ± 2.57 | 0.76 ± 0.06 |
| PCR | 14.80 ± 11.90 | 13.52 ± 15.55 | 1.22 ± 0.27 |
| Random Forest | 0.17 ± 0.11 | **1.80 ± 1.57** | 1.04 ± 0.17 |
| LightGBM | **0.17 ± 0.08** | 2.11 ± 0.97 | 0.99 ± 0.13 |
| LSTM | 23.79 ± 0.27 | 7.50 ± 0.79 | 6.57 ± 0.36 |
| MLP | 63.38 ± 11.12 | 607.46 ± 489.41 | 16.89 ± 1.57 |
| GRU | 23.78 ± 0.27 | 7.50 ± 0.80 | 6.48 ± 0.36 |

As batteries progress in their lifecycle, the variances in charging data distribution further intensify, thereby significantly compromising the model's capacity for generalization.

Conversely, tree-based models consistently exhibit robust performance across diverse datasets, maintaining low prediction errors, even within the MATR dataset. This robustness intimates the strong baseline that tree classifiers establish in SOH estimation tasks. However, deep learning models have yet to decisively surpass traditional methods in SOH tasks. On a majority of datasets, the errors stemming from deep learning models significantly exceed those from traditional models, indicating a substantial potential for enhancement in deep SOH modeling.

It's essential to note that current methods are yet to achieve full optimization, and the labels employed are approximations based on standard-workload conditions. Accurate labeling requires the implementation of a standardized workload, a methodology that contradicts real-world conditions. This discordance renders the collection of independent and identically distributed training samples, which utilize realistic workloads whilst retaining standard-workload labels, unfeasible. This inherent contradiction persists as a challenge that the field has yet to effectively surmount.

## C.3 STATE OF CHARGE ESTIMATION

Estimating the State of Charge (SOC) is contingent upon both the discharge capacity at a given moment and the SOH for the current cycle. Consequently, acquiring completely accurate SOC labels is as challenging as it is for SOH. In our methodology, we predict the SOC under the realistic workload of the battery. For this task, we employed datasets from CALCE, HNEI, HUST, MATR, RWTH, SNL, and UL_PUR. Feature extraction incorporates estimated current, voltage, and time information post-interpolation, alongside the charge and discharge capacity curves from preceding cycles. Labels represent the proportion of the battery's remaining capacity at a specific point in a cycle to the current full battery capacity, multiplied by 100. The specific definition is in Formula 2. The performance of the methods of SOC prediction task is displayed in Table 5.

Contrasting with the results for RUL and SOH prediction, SOC benchmark results reveal that LightGBM consistently surpasses other methodologies in the majority of tasks, thereby positioning tree models as the prevailing state-of-the-art for SOC prediction. However, all methods exhibit sub-

Table 5: Evaluation Result for SOC (State of Charge) task

| Method | Test RMSE | | | |
| | CALCE | HNEI | HUST | MATR |
|---|---|---|---|---|
| Linear Reg | $6.755 \pm 0.58$ | $7.197 \pm 0.21$ | $5.647 \pm 4.35$ | $2.698 \pm 0.25$ |
| Ridge Reg | $6.755 \pm 0.58$ | $7.197 \pm 0.21$ | $5.646 \pm 4.35$ | $2.698 \pm 0.25$ |
| PLSR | $8.780 \pm 1.03$ | $7.822 \pm 0.41$ | $5.113 \pm 0.83$ | $2.700 \pm 0.25$ |
| PCR | $6.756 \pm 0.58$ | $7.197 \pm 0.21$ | $5.647 \pm 4.35$ | $2.698 \pm 0.25$ |
| LightGBM | $\mathbf{1.714 \pm 1.02}$ | $\mathbf{0.763 \pm 0.02}$ | $\mathbf{0.284 \pm 0.04}$ | $\mathbf{0.824 \pm 0.58}$ |
| LSTM | $46.502 \pm 0.85$ | $44.318 \pm 0.58$ | $27.992 \pm 1.14$ | $35.868 \pm 1.08$ |
| MLP | $38.252 \pm 5.22$ | $16.192 \pm 5.57$ | $4.597 \pm 1.29$ | $4.570 \pm 2.57$ |
| GRU | $47.538 \pm 0.70$ | $45.568 \pm 0.26$ | $28.298 \pm 0.27$ | $35.927 \pm 1.07$ |
| Method | Test RMSE | | | |
| | RWTH | SNL | UL_PUR | |
| Linear Reg | $\mathbf{64.498 \pm 11.64}$ | $12.775 \pm 0.67$ | $2.484 \pm 0.19$ | |
| Ridge Reg | $64.500 \pm 11.64$ | $12.775 \pm 0.67$ | $2.483 \pm 0.19$ | |
| PLSR | $64.504 \pm 11.64$ | $14.320 \pm 2.01$ | $3.933 \pm 0.29$ | |
| PCR | $\mathbf{64.498 \pm 11.64}$ | $12.775 \pm 0.67$ | $2.485 \pm 0.19$ | |
| LightGBM | $313.757 \pm 63.76$ | $\mathbf{2.616 \pm 1.83}$ | $\mathbf{0.844 \pm 0.27}$ | |
| LSTM | $435.523 \pm 46.46$ | $27.919 \pm 0.34$ | $51.834 \pm 0.83$ | |
| MLP | $2224.028 \pm 3812.76$ | $13.922 \pm 3.73$ | $50.185 \pm 0.57$ | |
| GRU | $435.551 \pm 46.44$ | $27.903 \pm 0.37$ | $51.037 \pm 0.97$ | |

optimal performance on the RWTH dataset, suggesting that current approaches still grapple with adapting to the full spectrum of battery aging patterns. Additionally, linear models persistently outperform deep learning models, implying that both the input features and the network architecture necessitate further optimization to fully exploit the potential of deep learning models in SOC prediction tasks.

# D ABLATION STUDY

In this section, we present an ablation study of existing methods, focusing on the features and hyperparameters used in the RUL prediction task.

## D.1 FEATURE SPACE ABLATION

Figure 5 illustrates the predictive performance of statistical models on the MIX dataset using different features, where the Variance, Discharge, and Full features are all derived from (Severson et al., 2019).

The predictive performance of the statistical models on the MIX dataset, using various features, offers distinct insights. The Variance feature, acting as a reliable baseline, provides consistent results across all models. Conversely, the Discharge feature demonstrates weaker performance with linear regression, PCR, and PLSR due to its non-linear characteristics in the MIX dataset, which includes diverse batteries and aging patterns. The Full feature displays a strong linear relationship with RUL, performing well across all models. Notably, both the Gaussian Process and Random Forest models yield effective results with the Discharge feature, underscoring its predictive prowess for RUL. The raw Qd(Vd) feature also performs commendably, demonstrating the models' capacity to learn directly from low-level data. However, a significant performance gap exists between the Qd(Vd) and Full features when using Random Forest, indicating a considerable scope for improvement in the models' ability to autonomously extract effective features from raw inputs.

## D.2 HYPERPARAMETER ABLATION

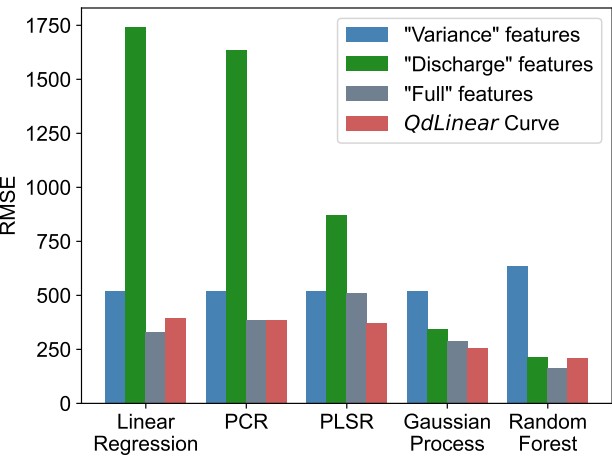

Figure 5: Feature space ablations. The "variance", "discharge" and "full" features are designed by domain experts to capture the degradation pattern of LFP/graphite cells. The "Variance" feature refers to the Log Variance of $\Delta Q_{100-10}(V)$ during the discharge process. The "Discharge" feature encompasses multiple features extracted from the discharge process. The "Full" feature includes features extracted from both the charging and discharging processes. *QdLinear* feature is obtained by linear interpolation of discharge capacity with respect to voltage.

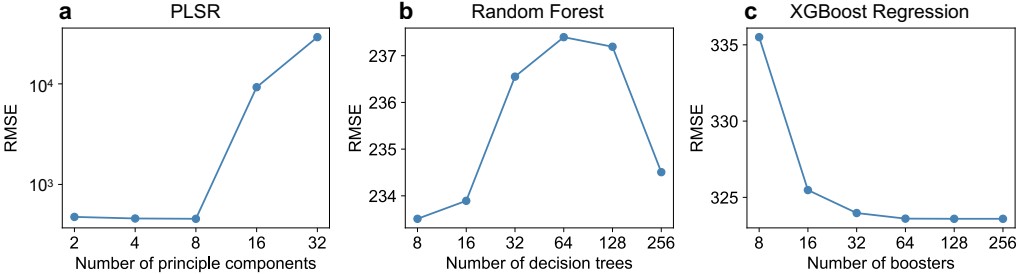

Figure 6: Hyper parameter analysis for traditional statistical models.

**Hyperparameter analysis for statistical models.** Figure 6 displays the hyperparameter analysis of traditional statistical methods. Figure 6a of the figure shows the performance of PLSR with varying numbers of principal components. PLSR exhibits a trend towards overfitting with an increasing number of principal components, necessitating cross-validation to determine the optimal number of components.

Figure 6b illustrates the impact of the number of decision trees on the performance of Random Forest. Random Forest shows stable effectiveness, indicating a preference for a moderate number of decision trees to balance performance and computational efficiency.

Figure 6c shows the performance change of XGBoost with the increasing number of boosters. An increase in the number of boosters improves effectiveness, with a convergence point observed beyond 64 boosters. This suggests the feasibility of employing a higher number of boosters while maintaining computational practicality.

**Hyperparameter analysis for deep models.** Figure 7 illustrates the impact of hidden dimensions on the performance of deep models. In Figure 7a, an increase in hidden dimensions results in higher variance without a significant change in mean prediction value, suggesting reduced robustness for larger dimensions in the `MIX` data source.

Figure 7b demonstrates that CNN require an optimal hidden dimension size, as extremes in size lead to high variance; a dimension of 16 is found to be most effective.

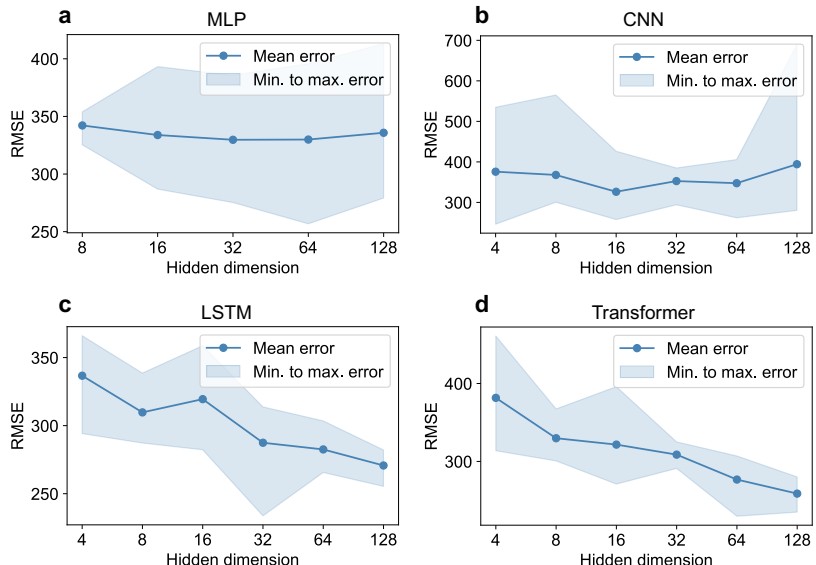

Figure 7: Hyper parameter analysis for deep models.

Figures 7c and 7d showcase the ablation results for LSTM and Transformer models. For sequence modeling approaches, such as LSTM and Transformer models, increased model capacity enhances performance, indicating the benefit of using larger model capacities in practical battery modeling scenarios.

# E    FLEXIBLE EXTENSIONS BASED ON BATTERYML

## E.1    PIPELINE INTERFACE FOR EFFICIENT REPLICATION

Code 1: An example configuration file for "Variance" model on `MATR1`.

```
train_test_split:
    name: 'MATRPrimaryTestTrainTestSplitter'
    cell_data_path: 'data/processed/MATR'
feature:
    name: 'VarianceModelFeatureExtractor'
    interp_dims: 1000
    critical_cycles:
        - 2
        - 9
        - 99
    use_precalculated_qdlin: True
feature_transformation:
    name: 'ZScoreDataTransformation'
label:
    name: 'RULLabelAnnotator'
label_transformation:
    name: 'SequentialDataTransformation'
    transformations:
        - name: 'LogScaleDataTransformation'
        - name: 'ZScoreDataTransformation'
model:
    name: 'LinearRegressionRULPredictor'
```

Almost all modules in BatteryML can be configured using the configuration file. Code 1 demonstrates an example that specifies "Variance" model on `MATR1` dataset. Here we introduce these components in detail.

**Module** `train_test_split`.   This module is responsible for dividing the data into training and testing sets. Developers can specify the or implement `Splitter` classes and specify in the "name" field. We provided predefined `Splitter` classes for convenient reproduction of existing methods. Please refer to our implementation in `batteryml/train_test_split` for details.

**Module** `feature`.  This module controls the feature extraction process, where we not only provide full freedom to users for customization, but also implemented typical features such as Variance, Discharge, and Full model based on $\Delta Q_{100-10}(V)$. The predefined features are organized in `batteryml/feature`.

**Module** `label`.  This module calculates the training target for each cell (e.g., for RUL prediction) or cycle (e.g., for SOH prediction). BatteryML will take care of the labeling process according to the specifications provided. In the given example, the `RULLabelAnnotator` will automatically process each cell and annotate their cycle life according to the specified end-of-life percentage. The implementations are located in `batteryml/label`.

**Module** `feature_transformation` **and** `label_transformation`.  These two module is responsible for post-processing features and labels before training. Examples include normalization and data augmentation. In addition to common transformations, we also implemented a sequential transformation wrapper that allows flexible compositions of transformations. For example, in Code 1 we transforms the label by first converting to log scale and then applying z-score normalization. The corresponding code path is `batteryml/data/transformation`.

**Module** `model`.  This module allows users to conveniently define models and their parameters. We provide common models in BatteryML for reproducing existing studies and serving as references for implementing custom models. The model definitions are located in `batteryml/models`.

Combining these modules together, researchers can employ the 'Pipeline' of BatteryML to conduct experiments efficiently, as demonstrated in Code 2. Please refer to the documentation for more examples.

Code 2: Example usage of the `Pipeline` API for "Variance" model training and evaluation.

```
from batteryml.pipeline import Pipeline
from batteryml.visualization.plot_helper import import plot_result

# Create a pipeline with a config file by specifying the config_path and workspace.
# Developers need to modify the data, feature, model and other related settings in the config
    file in advance.
pipeline = Pipeline(config_path='configs/baselines/sklearn/variance_model/matr_1.yaml',
                    workspace='workspaces')
# Model training. The training process will follow the config file, and the checkpoints will
    be saved to workspace.
model, dataset = pipeline.train(device='cuda')

# Also, developers can use previously trained models for evaluation. The result will be saved
    to workspace.
pipeline.evaluate(checkpoint='<your checkpoint path>')

# plot result
prediction = model.predict(dataset, data_type='test').to('cpu')
ground_truth = dataset.test_data.label.to('cpu')
plot_result(ground_truth, prediction)
```

## E.2  Unified Data Format for Public and Custom Data

BatteryML abstracts a general representation `BatteryData` for the diverse battery data records in existing studies. This abstraction lies at the core of BatteryML, which organizes the arbitrary form of battery data into a unified format, covering most public datasets and the output formats of typical battery cycling test equipment. Here we use MATR data source as an example for demonstration. As shown in Table 6, each `BatteryData` represents a battery cell. For each cell, meta information such as cathode material, anode material, etc., as well as charge-discharge cycle data and charge-discharge protocols are recorded. We assume that each cell is related to an array of charge-discharge cycles, which we store as a list of cycling records, as shown in Table 7. Each cycle comprises time-series records that include electrical signals such as voltage and current, and may also incorporate other parameters like temperature, resistance, mass, etc. Table 8 demonstrates the data entry method for charge-discharge protocols using a discharge protocol as an example. For instance, in the MATR dataset, the discharge is constant current, so the rate_in_C is fixed at 4C, the starting capacity is 1, the ending capacity is 0, and the cycle is continuous. BatteryML also provides a command line interface to conveniently organize existing public datasets into `BatteryData`, as shown in Code 3. Currently BatteryML supports cell-level learning of batteries for degradation modeling or battery optimization. In the future, BatteryML will accommodate a wider range of real-world applications, including higher-level battery packages such as BMS records of electric vehicles and lower-level data such as half-cells and material properties. BatteryML will continually foster data exchange in

Table 6: Example of the battery meta information of the unified data format.

| Attribute | Sample |
|---|---|
| cell_id | MATR_b1c1 |
| form_factor | cylindrical_18650 |
| anode_material | graphite |
| cathode_material | LFP |
| electrolyte_material | None |
| nominal_capacity_in_Ah | 1.1 |
| depth_of_charge | 1.0 |
| depth_of_discharge | 1.0 |
| already_spent_cycles | 0 |
| max_voltage_limit_in_V | 3.5 |
| min_voltage_limit_in_V | 2.0 |
| max_current_limit_in_A | 4.0 |
| min_current_limit_in_A | 0.0 |
| description | cell data of MATR dataset |
| cycle_data | list of CycleData |
| charge_protocol | list of CyclingProtocol |
| discharge_protocol | list of CyclingProtocol |

Table 7: Example of the cycling records of the unified data format. The default field primarily includes electrical data due to its high accessibility. Other types of data can be added as needed.

| Attribute | Sample |
|---|---|
| cycle_number | 1 |
| voltage_in_V | [2.0220623, 2.0347204, 2.0466299...] |
| current_in_A | [0.0, 0.216028, 0.360339...] |
| charge_capacity_in_Ah | [0.0, 1.0401919e-06, 1.0401919e-06...] |
| discharge_capacity_in_Ah | [0.0, 6.3991529e-10, 6.3991529e-10...] |
| time_in_s | [0.0, 0.002659, 0.003086...] |
| temperature_in_C | [31.376623, 31.376623, 31.376623...] |
| internal_resistance_in_ohm | 0.017038831 |

Table 8: Example of the cycling protocol specification of the unified data format. In practice, a battery may employ a sequence of such cycling protocols to get fully charged or discharged.

| Attribute | Sample |
|---|---|
| rate_in_C | 4.0 |
| current_in_A | None |
| voltage_in_V | None |
| power_in_W | None |
| start_voltage_in_V | None |
| start_soc | 1.0 |
| end_voltage_in_V | None |
| end_soc | 0.0 |

the battery domain, reducing the barriers to data-driven battery modeling, thereby paving the way for future scientific research.

Code 3: Illustrative Usage for Download Battery Data and Convert Data Format.

```
pip install -r requirements.txt
pip install .

batteryml download MATR /path/to/save/raw/data
batteryml preprocess MATR /path/to/save/raw/data /path/to/save/processed/data
```

E.3  FEATURE INTERFACE FOR DOMAIN KNOWLEDGE INCORPORATION

BatteryML provide a general interface for feature extraction, which takes the `BatteryData` for each cell as input and output extracted features in `torch.Tensor`. Note that the output features of the same cell may contain multiple training instances depending on the task. For example, the feature extractor may output one feature for each cycle of the same cell.

To develop custom feature extractors, developers simply need to implement the `process_cell` function, which is designed to handle a specific cell. BatteryML efficiently manages the iteration through the batteries and organizes the data collation into structured datasets. Code 4 demonstrates a practical example that calculates the Coulombic efficiency as input features.

Code 4: Example of adding feature Coulombic Efficiency as input.

```python
# batteryml/feature/new_feature.py
import torch

from typing import List

from src.builders import FEATURE_EXTRACTORS
from src.data.battery_data import BatteryData
from src.feature.base import BaseFeatureExtractor
@FEATURE_EXTRACTORS.register()
class NewFeatureExtractor(BaseFeatureExtractor):
    def __init__(self,
                 min_cycle_index: int = 0,
                 max_cycle_index: int = 99):
        self.min_cycle_index = min_cycle_index
        self.max_cycle_index = max_cycle_index

    def process_cell(self, cell_data: BatteryData) -> torch.Tensor:
        coulombic_efficiencies = []
        # Loop over each cycle in the cell data
        for cycle_index, cycle_data in enumerate(cell_data.cycle_data):
            if self.min_cycle_index <= cycle_index <= self.max_cycle_index:
                # Compute the coulombic efficiency for the current cycle
                ce = f_cycle_coulombic_efficiency(cycle_data.discharge_capacity_in_Ah,
                    cycle_data.charge_capacity_in_Ah)
                coulombic_efficiencies.append(ce)
        print(coulombic_efficiencies)
        coulombic_efficiencies = torch.FloatTensor(coulombic_efficiencies)
        feature = torch.tensor([torch.mean(coulombic_efficiencies), torch.std(
            coulombic_efficiencies), torch.var(coulombic_efficiencies)])
        print(feature)
        # Replace any NaN or infinite values in the feature tensor with zero
        feature[torch.isnan(feature) | torch.isinf(feature)] = 0.

        # Return the final feature tensor
        return feature

def f_cycle_coulombic_efficiency(Q_d, Q_c):
    return Q_d[-1] / (Q_c[-1] + 1e-5)
```

The `FEATURE_EXTRACTORS` registry manager will add the custom feature extractor into BatteryML. After we add this class into the import stack of BatteryML, coulombic efficiency feature can be used in configuration files for training, as shown in Code 5.

Code 5: Example configuration file that utilizes the coulombic effiency feature for training and evaluation.

```yaml
model:
    name: 'LinearRegressionRULPredictor'
train_test_split:
    name: 'MATRPrimaryTestTrainTestSplitter'
    cell_data_path: 'data/processed/MATR'
feature:
    name: 'NewFeatureExtractor'
    min_cycle_index: 0
    max_cycle_index: 99
label:
    name: 'RULLabelAnnotator'
feature_transformation:
    name: 'ZScoreDataTransformation'
label_transformation:
    name: 'SequentialDataTransformation'
    transformations:
```

```
        - name: 'LogScaleDataTransformation'
        - name: 'ZScoreDataTransformation'
```

## E.4 PREPROCESSING INTERFACE FOR CUSTOM DATA CLEANING AND AUGMENTATION

Data preprocessor in BatteryML is used for processing the feature before training and evaluation, whose input is the output of the feature extractors. Common usage of data preprocessors include data normalization and augmentation. The implementations are organized at `batteryml/data/transformation`. For each data processor, a `transform` and `inverse_transform` method is required. BatteryML also includes a sequential transformation API that allows users to use transformation compositions flexibly.

Code 6 showcases an example of custom data preprocessor, where the input data is normalized by the minimum and maximum values. Code 7 demonstrated an configuration example that uses this min-max-normalization feature for learning.

Code 6: Config Setting : Use the newly added preprocessor Min-Max normalization.

```python
# src/data/transformation/min_max.py
import torch
from src.builders import DATA_TRANSFORMATIONS
from src.data.transformation.base import BaseDataTransformation

@DATA_TRANSFORMATIONS.register()
class MinMaxDataTransformation(BaseDataTransformation):
    def __init__(self, base: float = None):
        self.min = None
        self.max = None

    def fit(self, data: torch.Tensor) -> torch.Tensor:
        self.min = torch.min(data)
        self.max = torch.max(data)

    def assert_fitted(self):
        assert self.min is not None, 'Transformation not fitted!'
        assert self.max is not None, 'Transformation not fitted!'

    @torch.no_grad()
    def transform(self, data: torch.Tensor) -> torch.Tensor:
        self.assert_fitted()
        data =  (data - self.min ) / (self.max  - self.min )
        return data

    @torch.no_grad()
    def inverse_transform(self, data: torch.Tensor) -> torch.Tensor:
        self.assert_fitted()
        data = data * (self.max  - self.min ) + self.min
        return data

    def to(self, device):
        self.min = self.min.to(device)
        self.max = self.max.to(device)
        return self
```

Code 7: Example configuration file that uses custom feature preprocessor for learning.

```yaml
model:
    name: 'LinearRegressionRULPredictor'
train_test_split:
    name: 'MATRPrimaryTestTrainTestSplitter'
    cell_data_path: 'data/processed/MATR'
feature:
    name: 'DischargeModelFeatureExtractor'
    interp_dims: 1000
    critical_cycles:
        - 2
        - 9
        - 99
    use_precalculated_qdlin: True
label:
    name: 'RULLabelAnnotator'
feature_transformation:
    name: 'MinMaxDataTransformation'
label_transformation:
    name: 'SequentialDataTransformation'
    transformations:
```

```
    – name: 'LogScaleDataTransformation'
    – name: 'MinMaxDataTransformation'
```

### E.5 Model Interface for Custom Battery Models

BatteryML provides a general model interface that follows the convention of 'scikit-learn' to support flexible model application to different learning tasks. Traditional statistical models such as linear models, Gaussian Process Regression, tree-based models, SVMs, are required to implement the `fit` and `predict` method for automatic learning process. Neural network models, on the other hand, merely needs to implement the forward logic. BatteryML encapsulates the training and evaluation process of the neural network into `fit` and `predict` methods to align with traditional statistical models for consistent behavior in the learning pipeline. Here we provide an example of using `LightGBM` for RUL prediction.

Since `lightgbm.LGBMRegressor` already supports `fit` and `predict` for predictions, we can directly inherit the `batteryml.model.SklearnModel` class and pass the hyperparameters of the model through configuration file. Code 8 demonstrates the example implementation for the RUL predictor using `LightGBM`. In this example, BatteryML will call the `fit` and `predict` method of `LightgbmRULPredictor.model` for the training and evaluation. Finally, the custom `LightGBM` model is registered to BatteryML after `@MODELS.register()` and imported during initialization.

Code 8: Implementation example of custom models: utilizing LightGBM as a case study. Hyperparameters are configured and passed through the configuration files.

```python
# batteryml/model/rul_predictor/lightgbm.py

from lightgbm import LGBMRegressor

from src.builders import MODELS
from src.models.sklearn_model import SklearnModel

@MODELS.register()
class LightgbmRULPredictor(SklearnModel):
    def __init__(self, *args, workspace: str = None, **kwargs):
        SklearnModel.__init__(self, workspace)
        self.model = LGBMRegressor(*args, **kwargs)
```

After defining the model, we can pass the hyperparameters of the RUL predictor through the configuration file, as shown in Code 9. The fields in the configuration file will be passed to the model as keyword arguments when initializing the predictor.

Code 9: Configuration example for custom LightGBM model.

```
model:
    name: 'LightgbmRULPredictor'
    boosting_type : 'gbdt'
    learning_rate : 0.001
    n_estimators : 200
    objective : 'regression'
train_test_split:
    name: 'MATRPrimaryTestTrainTestSplitter'
    cell_data_path: 'data/processed/MATR'
feature:
    name: 'VoltageCapacityMatrixFeatureExtractor'
    diff_base: 8
    max_cycle_index: 98
    cycles_to_keep: 98
    use_precalculated_qdlin: True
label:
    name: 'RULLabelAnnotator'
feature_transformation:
    name: 'ZScoreDataTransformation'
label_transformation:
    name: 'SequentialDataTransformation'
    transformations:
        – name: 'LogScaleDataTransformation'
        – name: 'ZScoreDataTransformation'
```

