# OpenReview forum: "BatteryML: An Open-source Platform for Machine Learning on Battery Degradation"
_ICLR.cc/2024/Conference — ICLR 2024 spotlight_

### Official Review · Reviewer_3yVC · 2023-10-28

**Soundness:** 3 good
**Presentation:** 3 good
**Contribution:** 3 good
**Rating:** 8
**Confidence:** 3

**Summary:**

This paper proposes an open-source toolkit for solving battery degradation problems, which is named as BatteryML. It not only provides datasets and feature engineering tools necessary for battery degradation research, but also provides a model library that can easily implement traditional and advanced battery degradation prediction models. By unifying data preprocessing, feature extraction, and model implementation, the proposed toolkit improves the practicality and efficiency of battery degradation research. At the same time, the open-source nature of BatteryML also promotes cooperation and contribution among experts from different fields, furthering the progress of battery technology.

**Strengths:**

1.	This is the first toolkit that bridges the battery degradation domain and the machine learning domain.
2.	The proposed toolkit unifies the data formats of diverse open datasets and comprises common models used in this domain, which is comprehensive.

**Weaknesses:**

1.	The technical and theoretical contribution of the toolkit itself is limited.

**Questions:**

no

---

> ### Author Response · Authors · 2023-11-18
> **Response to Reviewer 3yVC**
>
> Thank you for acknowledging the strengths of our work. In response to your concern about the toolkit's limited technical and theoretical contributions, we'd like to offer further insight into the platform's design and utility.
>
> Our toolkit, while not introducing groundbreaking machine learning algorithms or theories, plays a crucial role as a comprehensive and open-source resource for battery degradation modeling. The platform's core value lies in its ability to streamline the complex process of applying machine learning to battery degradation data, making it more accessible and efficient for researchers and engineers in the field.
>
> **Key features of the platform** include:
>
> - **Unified Interface**: The toolkit offers a cohesive interface for data preprocessing, feature extraction, and model building, significantly reducing the time and effort traditionally required in this research area.
> - **Reproducibility and Comparison**: By standardizing these processes, the platform facilitates reproducibility and enables more straightforward comparison of results across different studies, a crucial aspect in advancing scientific research.
> - **Extensibility**: Recognizing the dynamic nature of machine learning and battery science, we have designed the platform to be easily extendable. Users can integrate new data preprocessors, feature extractors, and models, as elaborated in the Appendix of our revised paper. This adaptability ensures that our platform remains relevant and useful as new techniques and research needs emerge.
>
> Looking ahead, we are committed to enriching the technical and theoretical aspects of the toolkit. We plan to actively collaborate with the data science and battery science communities to incorporate cutting-edge advancements and insights, thereby continuously enhancing the toolkit's capabilities and contributions to the field.

---

> > ### Comment · Reviewer_3yVC · 2023-11-21
> > **Response to authors**
> >
> > Thanks for your response.

---

### Official Review · Reviewer_9hBn · 2023-10-29

**Soundness:** 2 fair
**Presentation:** 3 good
**Contribution:** 3 good
**Rating:** 6
**Confidence:** 3

**Summary:**

This paper describes the current state of research in machine learning applied to battery degradation. The authors claim 3 challenges exist in current research:
1. Battery dataset are heterogenous in both their measurement techniques and data formats.
2. Feature extraction is difficult when applying ML techniques to battery data.
3. Adapting modern ML models to battery data cannot be done in a plug and play fashion.

To address these problems, the authors provide an open source platform called BatteryML which unifies datasets, provides tools for data preprocessing, feature extraction, model training, prediction and visualization. They also define the primary battery degradation prediction metrics including State of Charge (SOC), State of Health (SOH), and Remaining Useful Life (RUL). Finally, they evaluate multiple models (linear, neural nets, and random forests) on the task of predicting RUL separately on 7 different datasets as well as combinations of these datasets. They present test RMSE on the RUL prediction task and plots of different battery degradation metrics such as battery capacity vs number of charge cycles.

**Strengths:**

- The paper is written clearly, it lays out the current problems well and provides a good starting point for researchers to build their experiments/methods.

- The unified data format looks very useful for researchers to work with multiple datasets easily providing opportunities for future work in transfer learning and other areas.

- The set of standard feature extractors for battery data is also very useful for ML researchers without battery expertise to get started quickly in battery research.

- The paper provides a link to the git repository which includes more notebooks, plots, and other useful details.

**Weaknesses:**

- There is little discussion of which models/datasets performed best for the RUL task. Were there certain models which outperform others and if so why?

- While this paper provides a practical software tool for battery data research, it does not provide novel insights or modeling techniques that advance the state of the art in battery SOH or RUL prediction. While RMSE numbers for the RUL task are presented in table 2, the results are not discussed in the text nor are the advantages/disadvantages of each method. Additionally, the paper does not propose new models or methodology to improve the prediction performance on either the RUL or SOH tasks.

**Questions:**

- Table 2 has bolded results for the top set of rows (MATR1, MATR2, HUST, SNL), but not the second set. Could you bold the results in the second set of rows as well to highlight the best performing models on those datasets?

- Some table references in the appendix are not clickable.

---

> ### Author Response · Authors · 2023-11-18
> **Response to Reviewer 9hBn**
>
> Thank you for your thoughtful comments and suggestions. With our clarifications and revisions, we hope that we have addressed your concerns.
>
> **Response to Weakness -- Little Discussion on RUL Result**
>
> We appreciate your feedback calling for an expanded discussion on the performance of various models and datasets for the RUL task. Recognizing the importance of this analysis, we have broadened our discussion to include comprehensive performance analysis and explanations for the differing performances of the models.
>
> It is essential to note that **there is no singular method that consistently achieves competitive performance across all datasets**. Most methods display a lack of robustness, leading to substantial prediction errors in certain datasets, even when they demonstrate top accuracy in others.
>
> The Variance, Discharge, and Full models, which fit linear regression on features designed by domain experts for LiFePO4 (LFP) batteries, delivered commendable results on datasets such as *MATR2*, *HUST*, and *CLO*. These datasets utilize the same type of LFP batteries, suggesting that manually constructed features based on domain knowledge can be particularly effective for LFP batteries, demonstrating a strong linear relationship with the Remaining Useful Life (RUL).
>
> Traditional statistical models that utilize low-level *QdLinear* features yield significant results. Ridge Regression's consistent, albeit modest, performance across datasets confirms a strong linear relationship between the *QdLinear* feature and battery cycle life. However, its limitations on *HUST*, *CRUSH*, and *MIX* datasets underscore the challenges in maintaining this linear correlation under varying aging conditions. PCR and PLSR, which incorporate principal component analysis before linear regression, excel on *MATR1*, *CRUH*, and *SNL* datasets. This indicates a linear relationship of the QdLinear feature with battery life in different subspaces. Gaussian Process Regression (GPR) outperforms PCR and PLSR on the *MIX* dataset, due to its non-linear fitting ability, which is critical for capturing diverse battery degradation patterns. XGBoost and Random Forest demonstrate strong performance on *CRUSH* and *MIX* datasets, underscoring the effectiveness of tree-based models in modeling complex battery aging patterns.
>
> Neural network models, with the *QdLinear* feature, showed potential and instability. While their best performance under different seed settings might surpass some linear models (see the table below for CNN on *MATR1*), they also demonstrated a large variance, indicating poor model stability. Among them, the Transformer model exhibited better stability and achieved the best result on the mixed dataset *CRUSH*. This suggests that neural network models have significant potential in RUL prediction tasks, but the challenge of improving model stability persists.
>
> ||seed 0 | seed 1| seed 2| seed 3| seed 4| seed 5| seed 6| seed 7| seed 8| seed 9|
> | ----- | ----- | ----- | ----- | ----- | ----- | ----- | ----- | ----- | ----- | ----- |
> |CNN | 76 | 67 | 64 | 74| 60 | 82 | 65 | 79| 367 | 78 |
>
> In the revised version of the paper, we have provided a more detailed analysis of the RUL, SOH, and SOC results. This includes a comparative evaluation of diverse models on all battery datasets. Our aim with this enriched discussion is to illuminate the performance of each model under varying conditions, clearly highlighting their respective strengths and limitations.
>
>
> **Response to Weakness --  Not Propose New Models or Methodology**
>
> We appreciate your feedback regarding the lack of contributions on the model aspect. Our primary objective is to establish a robust foundation of classical baselines and allow in-depth analyses, which we believe will catalyze future research and innovations in battery degradation modeling. By furnishing researchers with a diverse array of datasets and a thorough benchmark for existing models, our tool aims to foster further exploration and development in the field. We envision our work as a stepping stone, enabling the community to build upon these foundational aspects and contribute novel approaches and methodologies in battery SOC, SOH, RUL prediction and beyond.
>
>
> **Response to Question 1 & Question 2**
>
> Thank you for pointing out the displaying issue in Table 2. We have adjusted this part in the revised version to ensure that the top-performing models in the different cases are all highlighted in bold.

---

> > ### Comment · Reviewer_9hBn · 2023-11-22
> > **Response to Authors**
> >
> > In light of the convincing additional analyses and discussion provided by authors, the reviewer will raise the score to 6.

---

### Official Review · Reviewer_ZTeZ · 2023-10-31

**Soundness:** 3 good
**Presentation:** 4 excellent
**Contribution:** 4 excellent
**Rating:** 8
**Confidence:** 3

**Summary:**

The paper proposes a software framework titled BatteryML that provides a unified data representation for the multitude of publicly available datasets, each with its own unique set of variables, goals and representation schemes.
In addition to providing an opportunity for battery science and ML experts to collaborate, the platform is meant to encourage the development of algorithms that are less specific to certain data collection schemes and more generally applicable.
Finally, with an array of tools for feature extraction, data preprocessing and access to more modern machine learning models, BatteryML seeks to collectively advance the state of battery research.

**Strengths:**

- The quality of writing and clarity go a long way to contextualizing how and why the proposed framework is important.
- The choice of long and detailed examples to highlight different portions of the software framework adds depth to the discussions about the framework.
- Figure 1 gives a concise overview of the entire framework, that is quickly digestible and informative.

**Weaknesses:**

- While BatteryML puts a lot of effort into standardizing publicly available datasets and making them accessible to work with, for the wider community, I'm curious how much of the detailed battery science gets communicated to the end-user, or whether the layer of abstraction necessary to standardize the framework obfuscates the nuances to a large degree. The reason this is important is to enable the development of models that actively support/engage with physical phenomena in the correct fashion (physically plausible explanations).
- There seems to be a mismatch between the keywords used in Figure 1 (Normalization) and subsequent descriptions, which use Data Preprocessing.
- SOC seems to be used prior to being explicitly defined (1st occurrence in Pg. 2, Contributions while definition is in Section 3.3)

**Questions:**

- Could the authors clarify how much communication, about the details of battery science and the nuances of various electro-chemical processes involved in battery science is effectively relayed to the end-user, to help formulate models that match physical phenomena? In case this is limited, could the authors justify how the framework could help machine learning researchers effectively collaborate with battery science researchers?
- Could the authors clarify which among Normalization or Data preprocessing is the correct nomenclature, and update the manuscript to maintain consistency?
- In addition, please make sure that all acronyms remain well defined prior to their usage.
- The addition of an example detailing a custom data preprocessing/feature extraction technique would highlight the flexibility of the framework even further.

---

> ### Author Response · Authors · 2023-11-18
> **Response to Reviewer ZTeZ**
>
> **Response to Weakness 1 and Question 1**
>
> Thanks so much for pointing out such insightful and visionary comments. We hope the following clarifications and explanations can help to address your concerns.
>
> Firstly, we would like to clarify that our unified data representation does not obstruct the communication between the nuances of battery science (e.g., electro-chemical processes) and the development of machine learning methods. On the contrary, this unified data layer is indispensable to facilitating such communication as it alleviates the burden of handling various data sources, thereby forming the data foundation for both training and evaluation processes. Our tool includes various vital and interpretable features such as incremental capacity, differential capacity, and coulomb efficiency, etc. These classical features, which incorporate essential physical insights about detailed electro-chemical processes, can be easily derived from our unified data representation. Furthermore, by accommodating different datasets, our tool enables rapid verification of the effectiveness of these features across diverse data scenarios.
>
> Additionally, we concur with your observation that there is significant room for enhancing the communication between battery science and data science. While there are many challenges to overcome, we envision BatteryML serving as a platform to facilitate these interactions and drive co-innovation among battery scientists and machine learning experts.
>
> One interesting case you have mentioned is to develop physics-informed methods, holding both data-driven flexibility and physically plausible explanations. Without doubt, BatteryML can effortlessly support the implementation of these methods due to its extensible interfaces across data, feature, and model modules. However, developing such methods can be challenging and requires either battery researchers equipped with professional ML skills or data scientists with a deep understanding of electro-chemical processes. We believe BatteryML can greatly contribute in this regard by handling most of the complex and repetitive work, facilitating focus on the modeling aspect, and providing a comprehensive benchmark for verification.
>
> Finally, your comments have inspired us to develop proactive mechanisms to foster communication between battery science and data science. For instance, we could incorporate visualizations or simulation examples to intuitively illustrate the details of battery science and inherent electro-chemical processes, which may inspire data scientists to incorporate proper insights into their models. Simultaneously, we could develop high-level interfaces to enable quick coding and verification for battery experts with limited programming skills.
>
> **Response to Questions 2, 3**
>
> Thanks for your proofreading. In the updated manuscript, the term `Data Preprocessing` will be consistently employed throughout the document. Additionally, we will ensure that all acronyms are clearly defined prior to their initial usage within the text.
>
> **Response to Question 4**
>
> As requested, we have included examples in the revised manuscript's appendix to illustrate the implementation of customized data preprocessing, feature engineering, and model development. For your convenience, an example of a customized data preprocessing step is detailed below.
>
> **Example of Adding New Data Preprocessor**
> Integrating a new data preprocessor into BatteryML involves two primary steps:
> 1. Define a new preprocessor class:
> The class definitions for data preprocessors are organized within  `batteryml/data/transformation`. To introduce a new data preprocessor, it is essential to implement interfaces such as `fit`, `transform`, and `inverse_transform`, closely resembles the design pattern of `scikit-learn`.
> 2. Import data preprocessor class in `__init__.py`:
> Our system employs automatic builders for the instantiation of BatteryML components. These builders register all declared classes in the registry for subsequent construction during the initial import of batteryml.
>
> Once these steps are completed, you can configure the parameters for feature and label transformations in the configuration file. BatteryML's `Pipeline` API is designed to streamline the learning and evaluation processes.

---

> > ### Comment · Reviewer_ZTeZ · 2023-12-01
> > **Response to authors**
> >
> > I appreciate the enthusiasm of the authors in their in depth response and the updates made in the revised manuscript.

---

### Official Review · Reviewer_g7qR · 2023-11-01

**Soundness:** 3 good
**Presentation:** 2 fair
**Contribution:** 3 good
**Rating:** 6
**Confidence:** 2

**Summary:**

This paper presents an open-sourced unified platform and framework for battery life research, including various modules such as configuration, data processing, feature extraction, modeling, and evaluation. Specifically, the authors integrated multiple data sources under a unified data representation, allowing comprehensive comparison of different approaches on a uniform set of evaluation criteria. This work enables ML researchers to propose more interesting and complicated battery science models.

**Strengths:**

**Originality:** To the best of my knowledge, this work fills the gap between battery science research and ML research. As stated in the paper, the lack of such a platform is a blocker for both the battery science community and the machine learning community.

**Clarity:** The paper is easy to follow in general, though it might be better to include more details on the methodology and framework with some discussions.

**Significance:** The authors have done significant enough work on data cleaning and processing, feature extraction, model implementation and evaluation, metrics reporting, etc.

**Weaknesses:**

One of the potential improvements of this work is to include more discussions on the compared algorithms. I think it will be really helpful for people using this framework to get to know the existing approaches better, and their advantages and limitations for different tasks, then making this work more impactful.

**Questions:**

Specifically, I could imagine the paper would be more convincing if more discussion and comparison are included. Some of the examples I could imagine include:
- Having more comparison and discussions between the methods from different categories, such as conventional methods vs tree-based methods vs NN-based methods.
- Within each category, it would be helpful to show the comparison as well.
- It would be helpful to have a more detailed ablation study on the hyperparameter choices, feature selections, etc.
- It could also be beneficial to include a transformer model as well for the NN-based method or explain why this is not a method considered in the paper.

---

> ### Author Response · Authors · 2023-11-18
> **Response to Reviewer g7qR(Part 1/4)**
>
> Thanks so much for all your thorough reviews, insightful comments, and many constructive suggestions, based on which we will further enrich the technical and experiment content of this paper. Besides, we hope the following responses can help answer your questions and address your concerns.
>
> **Response to Including More Discussions and Model Comparisons**
>
> Thank you for your valuable comments and suggestions. We appreciate your insight and have further elaborated on the comparisons and discussions of different methods in our revised paper to better address your concerns.
>
> Our comparative methods have been categorized into four types:
> - **Dummy Regressor**, a baseline using the mean of the training label for prediction.
> - **Linear models using handcrafted features**, such as Variance, Discharge, and Full Models, which employ linear regression on features devised by battery experts.
> - **Statistical models** using raw *Qd(Vd)* features, encompassing methods like Ridge Regression, PCR, PLSR, Gaussian Process, XGBoost, and Random Forest.
> - **Neural Network Models**, which include advanced models fitted to the *Qd(Vd)* feature, such as MLP, CNN, LSTM, and Transformer models.
>
> We initially compare these methods from the feature perspective and subsequently provide a detailed analysis of their strengths and weaknesses.
>
> **From a feature perspective**, Linear models, utilizing handcrafted features, have demonstrated satisfactory performance on datasets such as *MATR2*, *HUST*, and *CLO*, which solely consist of one battery type, LiFePO4 (LFP). This finding validates the efficacy of domain knowledge. However, these models appear to be less successful when applied to datasets that encompass a wider range of battery types and aging conditions, such as *CRUSH* and *MIX*. In these instances, models that are directly fitted on the *Qd(Vd)* curve have proven to be more effective than those using manually crafted features. This highlights a deficiency in domain-specific feature design and underscores the necessity for more versatile, generalizable features, emphasizing the potential advantages of automated representation learning.
>
> **Strengths and limitations of the three types of methods:**
> - Linear models using handcrafted features, such as the Discharge and Full model, offer relatively accurate predictions for LFP datasets. However, their performance diminishes on the *MIX* dataset, which features diverse aging conditions, due to the limited feature set and model capacity.
>
> - Traditional statistical models, capable of discerning non-linear patterns from low-level features such as *Qd(Vd)* curves, employ specific modeling mechanisms. These include the decision tree ensemble approach in Random Forests and the use of kernel functions in Gaussian Processes. Despite their robust performance on datasets like *CRUH*, *CRUSH*, and *MIX*, their efficacy decreases on datasets such as *MATR2* and *SNL*, characterized by a scarcity of training samples. This finding indicates that these statistical models require a larger volume of training data to effectively learn and represent meaningful insights in RUL task.
>
> - Neural networks, through automatic representation learning on low-level features, offer advancements, but face significant performance variations due to different random parameter initializations. For instance, our observations of CNN reveal its ability to make accurate predictions with many random seeds (as exemplified by the results on MATR1, shown below). However, certain seeds can lead to a surprising increase in error, causing significant regression error variations. This highlights both the potential benefits and challenges of applying neural networks to RUL prediction tasks.
>
> ||seed 0 | seed 1| seed 2| seed 3| seed 4| seed 5| seed 6| seed 7| seed 8| seed 9|
> | ----- | ----- | ----- | ----- | ----- | ----- | ----- | ----- | ----- | ----- | ----- |
> |CNN | 76 | 67 | 64 | 74| 60 | 82 | 65 | 79| 367 | 78 |
>
> In conclusion, our analysis indicates that no single method is universally optimal across all datasets. Each method exhibits unique strengths and weaknesses, suggesting the need for continued research in this domain.

---

> ### Author Response · Authors · 2023-11-18
> **Response to Reviewer g7qR(Part 2/4)**
>
> **Response to the Request for Detailed Comparison Discussion within Each Category**
>
> In response to your suggestion, we have included an in-depth discussion comparing the methods within each category, further elucidating the strengths and weaknesses of different group methodologies.
>
> **Linear Models with Hand-Crafted Features**: The Variance model, which uses the variance of the incremental *Qd(Vd)* curve as a scalar feature, displays moderate prediction accuracy across various datasets but generally falls short when compared to the Discharge and Full models. Both the Discharge and Full models exhibit variable effectiveness with significant errors on certain datasets, indicating a need for enhancements in feature design to improve the accuracy of linear model fitting.
>
> **Traditional Statistical Models**: These models achieve significant results by fitting on low-level features, such as raw *QdLinear* curves. Ridge Regression displays consistent, albeit modest, performance across datasets, affirming a strong linear correlation between *Qd(Vd)* feature and battery cycle life. However, its underperformance on datasets like *HUST*, *CRUSH*, and *MIX* underscores the challenges in linear correlations under varying aging conditions. PCR and PLSR, which apply principal component analysis and project cells into different subspaces before fitting a linear regression, excel in *MATR1*, *CRUH*, and *SNL* datasets. This suggests that the *Qd(Vd)* feature exhibits a linear relationship in different subspaces with the battery life. Gaussian Process Regression (GPR) outperforms PCR and PLSR on *MIX*, demonstrating the importance of GPR's non-linear fitting ability in accurately capturing the diverse degradation patterns of batteries with different electrode chemistry and operating conditions. XGBoost and Random Forest, exhibiting strong performance on *CRUSH* and *MIX* datasets, indicate that tree-based models' non-linear capabilities are adept at effectively modeling complex battery aging patterns.
>
> **Neural Network Models**: Neural network models, including Multilayer Perceptron (MLP), Convolutional Neural Network (CNN), Long Short-Term Memory (LSTM), and Transformer architectures, display significant performance variability. MLP performs well on datasets such as *SNL*, *CLO*, and *CRUH*, while CNN shows high sensitivity to initial conditions. LSTM, exhibiting less variability, proves its robustness and superior performance on the MATR2 dataset among the neural network methods. The Transformer architecture, emerging as the most robust among deep models, achieves the best results on the *CRUSH* dataset across all models. These findings underscore both the potential of neural networks and the necessity for further refinement in learning methods and architectural designs to maximize their potential and possibly outperform traditional models.

---

> ### Author Response · Authors · 2023-11-18
> **Response to Reviewer g7qR(Part 3/4)**
>
> **Response to the Request for Additional Ablation Studies.**
>
> In response to your recommendation, we have incorporated ablation studies that delve into feature space and hyperparameter analysis. Below, we provide a brief overview of the principal results.
>
> **Feature Ablations**
>
> | |"Variance" features | "Discharge" features | "Full" features| *Qd(Vd)* Curve|
> | ----- | ----- | ----- | ----- | ----- |
> |Linear Regression | 521 | 1743 | 331 | 395|
> |PCR | 521 | 1634 | 386 | 384 |
> |PLSR | 521 | 873 | 510 | 371 |
> |Gaussian Process | 521 | 341 | 287 | 257 |
> |Random Forest | 634 | 213 | 164 | 211 |
>
> - The "Variance" feature refers to the log variance of $∆Q_{100−10}(V)$ during the discharge process.
> - The "Discharge" feature encompasses multiple features extracted from the discharge process.
> - The "Full" feature includes features extracted from both the charging and discharging processes.
> - The *Qd(Vd)* Curve represents the *Q(V)* curve during the discharge process.
>
> The predictive performance of the statistical models on the *MIX* dataset, using various features, offers distinct insights. The Variance feature, acting as a reliable baseline, provides consistent results across all models. Conversely, the Discharge feature demonstrates weaker performance with linear regression, PCR, and PLSR due to its non-linear characteristics in the *MIX* dataset, which includes diverse batteries and aging patterns. The Full feature displays a strong linear relationship with RUL, performing well across all models. Notably, both the Gaussian Process and Random Forest models yield effective results with the Discharge feature, underscoring its predictive prowess for RUL. The raw *Qd(Vd)* feature also performs commendably, demonstrating the models' capacity to learn directly from low-level data. However, a significant performance gap exists between the *Qd(Vd)* and Full features when using Random Forest, indicating a considerable scope for improvement in the models' ability to autonomously extract effective features from raw inputs.
>
>
> **Hyperparameter Analysis for Statistical Models**
>
> |Numbers of principal components | 2| 4| 8|  16| 32|
> | ----- | ----- | -----| ----- | ----- | ----- |
> |RMSE of PLSR | 475 | 459 | 455 | 9262 | 29142 |
> |**Number of decision trees** | **8**|  **16** | **32**|  **64**| **128**| **256**|
> |RMSE of Random Forest | 233 | 234 | 236 | 237 | 237 | 234 |
> |**Number of boosters** | **8**|  **16** | **32**|  **64**| **128**| **256**|
> |RMSE of XGBoost | 336 | 325 | 324 | 323 | 323 | 323 |
>
> The hyperparameter analysis of traditional statistical methods offers valuable insights for practical applications. PLSR exhibits a trend towards overfitting with an increasing number of principal components, necessitating cross-validation to determine the optimal number of components. Random Forest shows stable effectiveness, indicating a preference for a moderate number of decision trees to balance performance and computational efficiency. In the case of XGBoost, an increase in the number of boosters improves effectiveness, with a convergence point observed beyond 64 boosters. This suggests the feasibility of employing a higher number of boosters while maintaining computational practicality.
>
> **Hyperparameter Analysis for Deep Neural Network Models**
>
> |Hidden dimension | 4| 8|  16| 32|  64| 128|256|512|
> | ----- | ----- | -----| ----- | ----- | ----- |  ----- | --- | --- |
> |MLP | - | 342 | 334 | 330 | 330 | 336 | - | - |
> |CNN | 375 | 368 | 326 | 353 | 347 | 394 | - | - |
> |LSTM | 337 | 310 | 319 | 287 | 282 | 270 | 248 | 257 |
> |Transformer | 382 | 330 | 322 | 309 |277 | 259 | 278 | 302 |
>
> The performance analysis of deep models reveals the influence of hidden dimensions on model robustness and accuracy. In the Multilayer Perceptron (MLP), an increase in hidden dimensions results in higher variance without a significant change in mean prediction value, suggesting reduced robustness for larger dimensions in the *MIX* data source. Convolutional Neural Networks (CNN) require an optimal hidden dimension size, as extremes in size lead to high variance; a dimension of 16 is found to be most effective. For sequence modeling approaches, such as LSTM and Transformer models, increased model capacity enhances performance, indicating the benefit of using larger model capacities in practical battery modeling scenarios.

---

> ### Author Response · Authors · 2023-11-18
> **Response to Reviewer g7qR(Part 4/4)**
>
> **Response to the Request for Incorporation of Transformer Model**
>
> We appreciate your suggestion to incorporate the Transformer model into our research. Initially, we had reservations about its use due to the data-scarce nature of battery degradation modeling, which could potentially impede the effective pre-training of the Transformer model. However, following your recommendation, we conducted experiments with the Transformer model and found it to be rather competitive. We present these findings below for your consideration.
>
> |Models | *MATR1* | *MATR2* | *HUST* | *SNL* |  *CLO* | *CRUH*| *CRUSH* | *MIX*|
> | ----- | ----- | -----| ----- | ----- | ----- |  ----- | ----- |  ----- |
> |MLP  | 149 ± 3 | 275 ± 27 |  459 ± 9  | 370 ± 81 |  146 ± 5  | 86 ± 4  | 448 ± 10 |  455 ± 37 |
> |CNN  | 102 ± 94 |  228 ± 104 |  465 ± 75  | 924 ± 267  | >1000 |  162 ± 116  | 354 ± 37  | 261 ± 38 |
> |LSTM  | 119 ± 11 |  219 ± 33 |  443 ± 29 |  539 ± 40  | 222 ± 12 |  97 ± 7  | 337 ± 19 |  266 ± 11 |
> |Transformer | 135 ± 13  | 364 ± 25| 391 ± 11| 424 ± 23| 187 ± 14 | 93 ± 5  |  **239 ± 9** |  259 ± 27|
>
>
> The Transformer model displayed consistent performance across various datasets, surpassing other neural network models. It notably achieved the best performance among all neural counterparts on the *CRUSH* dataset. Furthermore, our ablation study on neural network models shed light on the influence of increased hidden layer dimensions on prediction errors across all mixed datasets. These findings will be incorporated into the revised paper.

---

> ### Comment · Reviewer_g7qR · 2023-11-22
>
> I am thankful for the comprehensive response and the subsequent revisions made to the paper. In light of these updates, I have revised my evaluation and adjusted the score to 6.

---

### Author Response · Authors · 2023-11-21
**Revised Paper(Part 1/2)**

We thank the reviewers again for their insightful feedback!

We have made key amendments to the manuscript to better articulate our specific positioning and contributions. These include (a) a more comprehensive comparison and analysis of RUL, SOH, and SOC results, (b) the incorporation of transformers as a strong neural network baseline, (c) additional ablation studies on the features and hyperparameters analysis, (d) more example of data preprocessor, feature, and model modules, (e) a thorough description of the datasets used and (f.) Miscellaneous Clarifications. We are grateful for the reviewers' insightful queries and suggestions, which have substantially enhanced the quality of our paper.

In light of the extensive revisions and new material added to the appendix, we have reorganized it into four distinct sections for clarity and ease of navigation:
- A Benchmark Data Construction: Here, we provide an in-depth introduction to all data sources used in this study and detail the process of constructing our benchmark datasets.
- B Detailed Comparison Analysis: In this section, we benchmark and compare the performance of various machine learning models across three typical applications in battery aging modeling. It offers an exhaustive comparative analysis of different methodologies for degradation prediction.
- C Ablation Study: We present an ablation study of existing methods, with a focus on the features and hyperparameters used in the Remaining Useful Life (RUL) prediction task.
- D Flexible Extensions Based on BatteryML: This section thoroughly describes the utilization and extension of the BatteryML toolkit, providing guidance on how to effectively implement and adapt it for diverse applications.

**(a) Comparison and analysis of RUL, SOH and SOC results**

[Reviewers 1 and 3]:
- **Section 4.2 and New Appendix B.1** (Remaining useful life prediction): Our expanded analysis of RUL task delves into the strengths and weaknesses of methods within each model category. We find no universally superior method in battery modeling, with each model outperforming others in specific datasets and underperforming in others, indicating room for improvement in early battery life prediction accuracy. Linear models work well on LiFePO4 (LFP) battery datasets but struggle with the diverse conditions in the *MIX* dataset due to their limited feature set and model capacity. Traditional statistical models such as Random Forests and Gaussian Processes can detect non-linear patterns in low-level features like *Qd(Vd)* curves, but their performance diminishes with datasets like *MATR2* and *SNL* that have fewer training samples, pointing to a need for larger datasets. Neural network models, which use automatic representation learning on low-level features, show promise but also exhibit large performance variations with different random parameter initializations.

[Reviewers 1 and 3]:
- **Section 4.2 and New Appendix B.2** (State of Health estimation): We delve into the SOH task, outlining feature construction, label annotation, and model result analysis. While linear models are generally effective, they struggle with MATR cells due to diverse charging strategies. Tree-based models, however, exhibit robust performance across datasets, forming a strong SOH estimation baseline. In current state, deep learning models don't consistently surpass traditional methods, highlighting significant room for improvement.

[Reviewers 1 and 3]:
- **Section 4.2 and New Appendix B.3** (State of Charge estimation): We offer a comprehensive introduction and detailed comparative analysis to the SOC task. Our findings indicate that tree-based models, especially LightGBM, consistently surpass other methodologies in most SOC prediction tasks, establishing them as the current benchmark in the field. However, it's noteworthy that linear models continue to perform better than deep learning models. This observation underscores the need for additional research to effectively harness the capabilities of neural networks in the context of SOC estimation.

---

> ### Author Response · Authors · 2023-11-21
> **Revised Paper(Part 2/2)**
>
> **(b) Inclusion of Transformer Model in RUL Task:**
>
> [Reviewer 1]:
> - **Section 4.2** (Remaining useful life prediction) and Table 2: Per the suggestion of Reviewer 1, we have incorporated the Transformer model for comparison. The Transformer model showcased superior performance on various datasets among neural network baselines, especially on the *CRUSH* dataset. Our ablation study further illuminated the impact of increased hidden layer dimensions on prediction errors on the *MIX* dataset.
>
>
> **(c) Ablation Study on RUL Task Features and Hyperparameters:**
>
> [Reviewer 1]:
> - **Appendix C.1** (Feature space ablation): We studied the impact of different feature inputs on the performance of statistical models on the *MIX* dataset. Notably, the Random Forest model showed stable results across all features and delivered the best performance. The comparable results of *Qdlinear* feature and the "Full" features designed by experts, highlight the potential of data-driven methods.
>
> - **Appendix C.2** (Hyperparameter Ablation): We conducted an analysis focused on the selection of hyperparameters for both traditional statistical models and neural networks. Our findings reveal that tree-based models exhibit robustness regarding the number of decision trees utilized in their ensemble. LSTM and Transformer models show resilience to variations in hidden dimensions. However, CNN and MLP models display significant performance variances, underscoring the necessity of careful hyperparameter selection to achieve good results.
>
>
> **(d) Example of data preprocessor, feature, and model modules:**
>
> [Reviewers 2 and 4]:
>
> - **New Appendix D.3**: We illustrate how to add new features to our BatteryML platform in a simple, two-step process.
> - **New Appendix D.4**: We demonstrate how to add data preprocessing methods to BatteryML.
> - **New Appendix D.5**: We provide practical examples to show how to integrate new models into the BatteryML framework.
>
> All these extensions are seamlessly managed by BatteryML and can be effortlessly integrated into the existing pipeline.
>
> **(e) Benchmark dataset description**
>
> [Reviewer 4]:
>
> - **New Appendix A**: We offered a thorough description of the public data employed in our experiments. This includes detailed information about the data sources and how we construct the benchmark datasets for this study. This comprehensive overview ensures transparency and reproducibility in our experimental approach.
>
> **(f.) Miscellaneous Clarifications**
>
> [Reviewer 2]:
>
> - **Nomenclature and format**: The term Data Preprocessing was consistently employed throughout the document.  Additionally, all acronyms are clearly defined prior to their initial usage within the text.
>
> [Reviewer 3]:
>
> - **Format**: We have adjusted this part in the revised version to ensure that the top-performing models in the different cases are all highlighted in bold. Additionally, we fixed the issue of non-clickable table references in the appendix.
>
> With our clarifications and revisions, we hope that we have addressed the reviewers' concerns. Thank you for your kind consideration.

---

### Meta-Review · Area_Chair_UgXZ · 2023-12-06

**Metareview:**

The paper addresses an important application domain: ML-based prediction of remaining useful life (RUL) in batteries. In this regard, the authors introduce BatteryML, an open-source platform that battery researchers can leverage to benchmark existing (and future) prediction methods on a variety of datasets, potentially leading to unified standards for data formats and evaluation criteria.

This is a well-executed project, and reviewers unanimously appreciated the clarity of the paper, quality of the work, and potential for future impact in this application domain. The authors are advised to take into account the reviewers' comments while preparing revisions.

**Justification For Why Not Higher Score:**

While the BatteryML platform can benefit from advertisement among the AI/ML community, the contribution is not technically deep enough to merit a full-length oral.

**Justification For Why Not Lower Score:**

The BatteryML platform may spur further research in battery health management as well as serve as a rich source of datasets, so may benefit from advertisement among the ICLR audience (via a spotlight).

---

### Decision · Program_Chairs · 2024-01-16

Accept (spotlight)